# STOCHASTIC MULTI-PERSON 3D MOTION FORECASTING

**Sirui Xu    Yu-Xiong Wang*    Liang-Yan Gui***
University of Illinois at Urbana-Champaign
{siruixu2, yxw, lgui}@illinois.edu
https://sirui-xu.github.io/DuMMF

## ABSTRACT

This paper aims to deal with the ignored real-world complexities in prior work on human motion forecasting, emphasizing the social properties of multi-person motion, the diversity of motion and social interactions, and the complexity of articulated motion. To this end, we introduce a novel task of *stochastic multi-person 3D motion forecasting*. We propose a *dual*-level generative modeling framework that separately models independent individual motion at the *local* level and social interactions at the *global* level. Notably, this dual-level modeling mechanism can be achieved within a shared generative model, through introducing *learnable latent codes* that represent intents of future motion and switching the codes' modes of operation at different levels. Our framework is general; we instantiate it with different generative models, including generative adversarial networks and diffusion models, and various multi-person forecasting models. Extensive experiments on CMU-Mocap, MuPoTS-3D, and SoMoF benchmarks show that our approach produces diverse and accurate multi-person predictions, significantly outperforming the state of the art.

## 1 INTRODUCTION

One of the hallmarks of human intelligence is the ability to predict the evolution of the physical world over time given historical information. For example, humans naturally anticipate the flow of people in public areas, react, and plan their own behavior based on social rules, such as avoiding collisions. Effective forecasting of human motion has thus become a crucial task in computer vision and robotics, *e.g.*, in autonomous driving (Paden et al., 2016) and robot navigation (Rudenko et al., 2018). This task, however, is challenging. First, human motion is structured with respect to both body physics and social norms, and is highly dependent on the surrounding environment and its changes. Second, human motion is inherently uncertain and multi-modal, especially over long time horizons.

Previous work on human motion forecasting often focuses on simplified scenarios. Perhaps the most widely adopted setting is on stochastic local motion prediction of a single person (Mao et al., 2021; Yuan & Kitani, 2020), which ignores human interactions with the environment and other people in the environment. Another related task is deterministic multi-person motion forecasting (Wang et al., 2021b; Adeli et al., 2020; 2021; Guo et al., 2022). However, it does not take into account the diversity of individual movements and social interactions. In addition, stochastic forecasting of human trajectories in crowds (Alahi et al., 2014) has shown progress in modeling social interactions, *e.g.*, with the use of attention models (Kosaraju et al., 2019; Vemula et al., 2018; Zhang et al., 2019) and spatial-temporal graph models (Huang et al., 2019; Ivanovic & Pavone, 2019; Salzmann et al., 2020; Yu et al., 2020). Nevertheless, this task only considers motion and interactions at the trajectory level. Modeling articulated 3D poses involves richer human-like social interactions than trajectory forecasting which only needs to account for trajectory collisions.

To overcome these limitations, we introduce a novel task of *stochastic multi-person 3D motion forecasting*, aiming to jointly tackle the aforementioned aspects ignored in the previous work – the social properties of multi-person motion, the multi-modality of motion and social interactions, and

---

*Yu-Xiong Wang and Liang-Yan Gui contributed equally to this work.

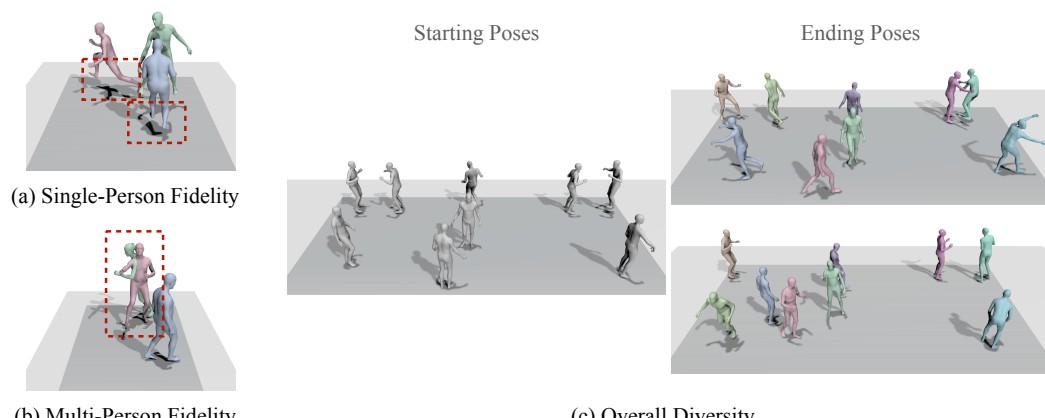

Figure 1: Illustration of the multifaceted challenges in the proposed task of stochastic multi-person 3D motion forecasting. (a) Single-person fidelity: for each person, the predicted pose and trajectory should be realistic and consistent with each other, *e.g.*, to avoid foot floating and skating. (b) Multi-person fidelity: multi-person motion in a scene inherently involves social interactions, *e.g.*, to avoid motion collisions. (c) Overall diversity: long-term human motion is uncertain and stochastic; we address this intrinsic multi-modality, while existing work (Wang et al., 2021b; Adeli et al., 2020; 2021; Guo et al., 2022) simplifies to deterministic prediction.

the complexity of articulated motion. As shown in Figure 1, this task requires the predicted multiple motion sequences of multi-person to satisfy the following conditions – (a) **single-person fidelity:** for example, all single-person motion should be continuous, and the articulated properties should be preserved under the physical rules; (b) **multi-person fidelity:** the predicted motion should be socially-aware, with the consideration of the interactions between predictions from different people; and (c) **overall diversity:** the movement of the human body should be as varied as possible, but within the constraints of conditions (a) and (b).

Due to the substantially increased complexity of our task, it becomes challenging to optimize all three objectives simultaneously. We observe simply extending existing work such as on deterministic motion forecasting cannot address the proposed task. This difficulty motivates us to adopt a *divide-and-conquer* strategy, together with the observation that single-person fidelity and multi-person fidelity can be viewed as relatively independent goals, while there is an inherent trade-off between fidelity and diversity. Therefore, we propose a ***Du**al-level generative modeling framework for **M**ulti-person **M**otion **F**orecasting* (DuMMF). At the *local* level, we model motion for different people *independently* under relaxed conditions, thus satisfying single-person fidelity and diversity. Meanwhile, at the *global* level, we model social interactions by considering the correlation between all motion, thereby further improving multi-person fidelity. Notably, this dual-level modeling mechanism can be achieved within a shared generative model, through simply switching the modes of operation of the *motion intent* codes (*i.e.*, latent codes of the generative model) at different levels. By optimizing these codes with level-specific objectives, we produce diverse and realistic multi-person predictions.

**Our contributions** can be summarized as follows. **(a)** To the best of our knowledge, we are the *first* to investigate the task of stochastic multi-person 3D motion forecasting. **(b)** We propose a simple yet effective dual-level learning framework to address this task. **(c)** We introduce discrete learnable social intents at dual levels to improve the realism and diversity of predictions. **(d)** Our framework is general and can be operationalized with various generative models, including generative adversarial networks and diffusion models, and different types of multi-person motion forecasting models. Notably, it can be generalized to challenging more-person (*e.g.*, 18-person) scenarios that are *unseen* during training.

## 2 RELATED WORK

**Stochastic Human Motion Forecasting.** There have been many advances in stochastic human motion forecasting, many of which (Walker et al., 2017; Yan et al., 2018; Barsoum et al., 2018) are based on the adaptation and improvement of deep generative models such as variational autoencoders

(VAEs) (Kingma & Welling, 2013), generative adversarial networks (GANs) (Goodfellow et al., 2014), normalizing flows (NFs) (Rezende & Mohamed, 2015), and diffusion models (Sohl-Dickstein et al., 2015; Song et al., 2020; Ho et al., 2020). Some recent approaches (Bhattacharyya et al., 2018; Dilokthanakul et al., 2016; Gurumurthy et al., 2017; Yuan & Kitani, 2019; 2020; Zhang et al., 2021; Mao et al., 2021; Xu et al., 2022b; Petrovich et al., 2022) emphasize on the promotion of diversity. Mao et al. (2021) sequentially generate the different parts of a pose for better controllability of diversity. Xu et al. (2022b) introduce learnable anchors in the latent space to guide the samples with sufficient diversity. Although these methods can predict very diverse human motion sequences, most of them are limited to local motion and ignore the global trajectory. Some of their produced motion sequences are actually unrealistic; in particular, incorporating global trajectories may result in severe foot skating. Predicting human motion under the constraint of scene context (Cao et al., 2020; Hassan et al., 2021; Zhang & Tang, 2022) has recently been explored, where the effect of global trajectories and scenes on human motion is considered. Instead of predicting single-person movement, our work focuses on diverse multi-person movements and social interactions.

**Multi-Person Forecasting.** So far, research on multi-person forecasting has mainly focused on global trajectory forecasting (Helbing & Molnar, 1995; Mehran et al., 2009; Pellegrini et al., 2009; Yamaguchi et al., 2011; Zhou et al., 2012; Alahi et al., 2014; 2016; Lee et al., 2017; Gupta et al., 2018; Sadeghian et al., 2019; Amirian et al., 2019; Kosaraju et al., 2019; Sun et al., 2020; Mangalam et al., 2020; Sun et al., 2021; Kothari et al., 2021; Tsao et al., 2022). Gupta et al. (2018) utilize a winner-takes-all loss (Rupprecht et al., 2017) with a socially-aware GAN. We also adopt these two loss functions in our dual-level modeling. Kothari et al. (2021) introduce discrete social anchors and revise the distribution of trajectories by the predefined anchors. Unlike their hand-crafted anchors, our discrete social intent codes are learnable components obtained through the dual-level optimization. Some recent attempts address *deterministic* multi-person 3D motion forecasting. Adeli et al. (2020; 2021) use additional contextual information to aid multi-person forecasting and propose a Social Motion Forecasting (SoMoF) Benchmark. Guo et al. (2022) utilize cross-attention for two-person motion forecasting, while Wang et al. (2021b) introduce a multi-range transformer architecture that can be generalized to more persons. Please refer to Sec. B of the Appendix for additional discussion.

## 3 METHODOLOGY

In this section, we explain the proposed dual-level generative modeling framework (DuMMF) for our task. As illustrated in Figure 2, our key insight is to *decouple the modeling of independent individual movements at the local level and social interactions at the global level* (Sec. 3.1). Within a shared forecasting model, we achieve this by (a) introducing *learnable* latent codes that represent intents of future movement (Sec. 3.2), (b) switching the codes' modes of operation at different levels (Sec. 3.1 and Sec. 3.2), and (c) training with level-specific objectives (Sec. 3.3). Our framework is general and can be operationalized with different types of motion forecasting models, and we summarize the multi-person motion predictors used in this paper (Sec. 3.4).

**Problem Formulation.** We denote the input motion sequence of length $T_h$ for $N$ persons in a scene as $\{\mathbf{X}_n\}_{n=1}^N$, where $\mathbf{X}_n[t]$ is the pose of $n$-th person at time step $t$. We aim to predict $M$ future motion sequences of length $T_p$, denoted as $\{\{\widehat{\mathbf{Y}}_n^m\}_{n=1}^N\}_{m=1}^M$, where $\widehat{\mathbf{Y}}_n^m = [\widehat{\mathbf{Y}}_n^m[T_h+1],\dots,\widehat{\mathbf{Y}}_n^m[T_h+T_p]]$ is the $m$-th predicted motion of $n$-th person. We use the 3D coordinates to represent the absolute joint position of $V$ joints, hence $\forall n, m, t, \mathbf{X}_n[t], \widehat{\mathbf{Y}}_n^m[t] \in \mathbb{R}^{V \times 3}$. We assume to be given the ground truth motion of $N$ persons as $\{\mathbf{Y}_n\}_{n=1}^N$. Our goal is to forecast multiple realistic yet diverse future motion sequences, such that (a) all the $M$ predictions represent human-like motion, simultaneously satisfying *single-person fidelity* and *multi-person fidelity*; (b) the predictions are diverse (*overall diversity*); and (c) one of the predicted sequences is as close to the ground truth as possible.

### 3.1 DUAL-LEVEL STOCHASTIC MULTI-PERSON MOTION FORECASTING: OVERVIEW

**Basic Generative Modeling Framework.** Stochastic multi-person future motion can be modeled as a joint distribution using deep generative models (Goodfellow et al., 2014). Accordingly, we denote this joint distribution of the future motion of $N$ persons as $p(\mathbf{Y}_1, \mathbf{Y}_2, \dots, \mathbf{Y}_N | \mathbf{X}_1, \mathbf{X}_2, \dots, \mathbf{X}_N)$, where all future movements $\{\mathbf{Y}_n\}_{n=1}^N$ are conditioned on the past sequences $\{\mathbf{X}_n\}_{n=1}^N$ of all persons. Typically, we can use a latent code $\mathbf{z} \sim p(\mathbf{z})$ to reparameterize this joint distribution as $p(\{\mathbf{Y}_n\}_{n=1}^N | \{\mathbf{X}_n\}_{n=1}^N) = \int p(\{\mathbf{Y}_n\}_{n=1}^N | \mathbf{z}, \{\mathbf{X}_n\}_{n=1}^N) p(\mathbf{z}) \mathrm{d}\mathbf{z}$. Here, the latent code $\mathbf{z}$ can be inter-

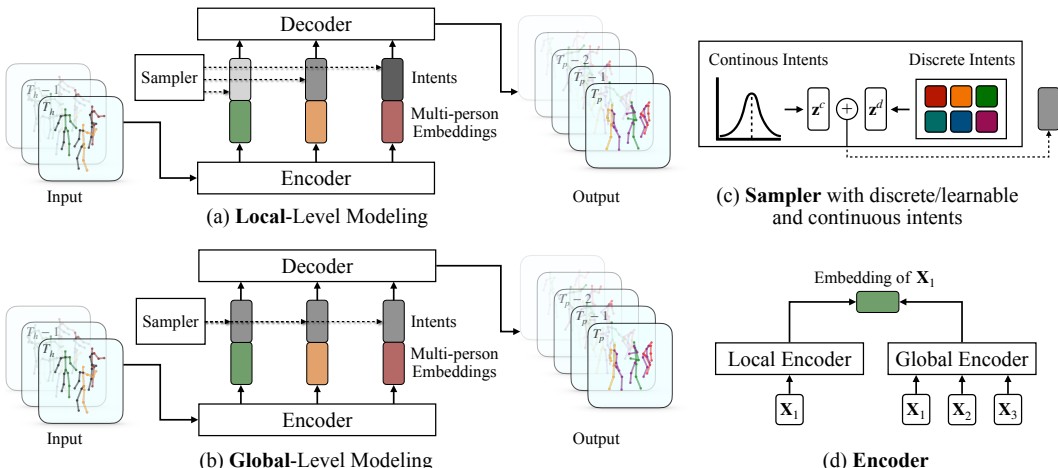

Figure 2: Overview of our proposed dual-level generative modeling with motion inputs of three persons as an illustration. (a) We combine encoded multi-person embeddings with *independent* intent codes at the local level of modeling individual motion. (b) Differently, the global level of modeling social interactions requires all latent codes to be the *same*. (c) The latent codes comprise both discrete intent codes, which are learned from the data and represented as a set, and continuous intent codes. (d) We abstract the encoder of a multi-person predictor as the combination of a local branch that encodes single-person motion and a global branch that encodes multi-person motions (see Table A of the Appendix for instantiations).

preted as a *social intent* that guides the future movements of multiple persons. With this social intent $z$ sampled from the given distribution $p(\mathbf{z})$, a deterministic neural network $\mathcal{G}_\theta$ with parameters $\theta$ can map the historical conditions $\{\mathbf{X}_n\}_{n=1}^N$ to a prediction $\{\widehat{\mathbf{Y}}_n\}_{n=1}^N$ under the latent distribution $p_\theta(\{\mathbf{Y}_n\}_{n=1}^N|\mathbf{z}, \{\mathbf{X}_n\}_{n=1}^N)$, which is formulated as

$$\mathbf{z} \sim p(\mathbf{z}), \ \{\widehat{\mathbf{Y}}_n\}_{n=1}^N = \mathcal{G}_\theta(\mathbf{z}, \{\mathbf{X}_n\}_{n=1}^N). \tag{1}$$

Given the complexity of our task, it becomes challenging to simultaneously ensure all objectives (*i.e.*, the fidelity of a single person, the fidelity of multiple persons, and the overall diversity). To overcome this difficulty, we introduce a dual-level modeling mechanism that explicitly decomposes the task objectives into local modeling of independent individual movements and global modeling of social interactions. Notably, we achieve this by simply switching the modes of operation for the latent codes $\mathbf{z}$ w.r.t. different levels of modeling, without any change to the model architecture $\mathcal{G}$.

**Local-Level Modeling: Individual Motion.** At this level, the generative model $\mathcal{G}_\theta$ models all human bodies as independent of each other, and we aim to improve the *overall diversity* and the *single-person fidelity*, alleviating problems such as predicting unrealistic poses. Here, the joint distribution of future human motions can be rewritten in the form of all single-person marginal distributions, *i.e.*, $p(\{\mathbf{Y}_n\}_{n=1}^N|\{\mathbf{X}_n\}_{n=1}^N) = \prod_{n=1}^N p(\mathbf{Y}_n|\mathbf{X}_n)$. To this end, as shown in Figure 2(a), we consider leveraging $N$ different *individual intents* $\mathbf{z}_1, \mathbf{z}_2, \dots, \mathbf{z}_N$ independently drawn from $p(\mathbf{z})$ to generate independent future movements, denoted as

$$\mathbf{z}_1, \dots, \mathbf{z}_n \sim p(\mathbf{z}), \ \{\widehat{\mathbf{Y}}_n\}_{n=1}^N = \mathcal{G}_\theta(\{\mathbf{z}_n\}_{n=1}^N, \{\mathbf{X}_n\}_{n=1}^N). \tag{2}$$

**Global-Level Modeling: Social Interactions.** Going beyond the local individual level, the generative model $\mathcal{G}_\theta$ at the global level takes into account the social behavior of multiple people to model their joint distribution. The goal is to further improve the *multi-person fidelity*, *e.g.*, promoting the overall accuracy. As illustrated in Figure 2(b), to maintain the network architecture $\mathcal{G}$ unchanged, we still use $N$ *individual intents* as input. However, different from the local level, we constrain these $N$ individual intents to be the same, representing *social intents* that stand for correlations between the intents of multiple persons. Formally, we have

$$\mathbf{z} \sim p(\mathbf{z}), \ \{\widehat{\mathbf{Y}}_n\}_{n=1}^N = \mathcal{G}_\theta(\{\mathbf{z}\}_{n=1}^N, \{\mathbf{X}_n\}_{n=1}^N). \tag{3}$$

Note that, without additional constraints, this dual-level modeling scheme by itself is not guaranteed to enforce the latent codes to behave in the designed manner. To this end, we introduce learnable latent intent codes $\mathbf{z}$ (Sec. 3.2), jointly optimize the codes $\mathbf{z}$ and the forecasting model $\mathcal{G}$ guided by the level-specific training objectives (Sec. 3.3).

## 3.2 DISCRETE LEARNABLE HUMAN INTENT CODES

Intuitively, an arbitrary, albeit identical, individual intent in Eq. 3 may not adequately lead to a valid social intent. We thus hypothesize that a social intent is formed when all individual intents are *the same and belong within some range of "options."* This can typically be achieved through discrete choice models (Aguirregabiria & Mira, 2010; Ryan & Gerard, 2003; Bhat et al., 2008; Leonardi, 1984) – an effective tool that predicts choices from a set of available options created by hand-crafted rules. Here, we formulate the correlation of multiple persons at the global level by using the same discrete code. However, the intent options for social interactions are more subtle and difficult to define manually than those in other applications such as for trajectories (Kothari et al., 2021). Therefore, we use a set of learnable codes $\mathbf{z}^d \in \{\mathbf{Z}^m\}_{m=1}^M$ to represent social intents, inspired by Xu et al. (2022b). However, we introduce different training strategies that are tailored to this new task (Sec. 3.3). Our motivations are: (a) Subject to physical constraints and social laws, the intents of future movements should share some deterministic patterns. For example, all intents should avoid imminent collisions anticipated from the history, even if these intents refer to different motions. We assume that such deterministic properties shared by social intents can be represented by a set of shareable codes learned directly from the data. (b) It will be easier for the predictor to identify and implement different levels of functionality by jointly optimizing discrete intents and the predictor. To further enhance the expressiveness of the codes, we retain the original continuous Gaussian noise $\mathbf{z}^c \sim p(\mathbf{z})$ of the generative model $\mathcal{G}_\theta$ and bundle the discrete intent with the noise to represent the final intent, as shown in Figure 2(c). Now the global-level modeling of social interactions in Eq. 3 is reformulated as

$$\mathbf{z}_1^c, \ldots, \mathbf{z}_n^c \sim p(z), \ \mathbf{z}^d \in \{\mathbf{Z}^m\}_{m=1}^M, \ \{\widehat{\mathbf{Y}}_n\}_{n=1}^N = \mathcal{G}_\theta(\{\mathbf{z}_n^c + \mathbf{z}^d\}_{n=1}^N, \{\mathbf{X}_n\}_{n=1}^N). \qquad (4)$$

And correspondingly, the local-level modeling of individual motion in Eq. 2 becomes

$$\mathbf{z}_1^c, \ldots, \mathbf{z}_n^c \sim p(z), \ \mathbf{z}_1^d, \ldots, \mathbf{z}_n^d \in \{\mathbf{Z}^m\}_{m=1}^M, \ \{\widehat{\mathbf{Y}}_n\}_{n=1}^N = \mathcal{G}_\theta(\{\mathbf{z}_n^c + \mathbf{z}_n^d\}_{n=1}^N, \{\mathbf{X}_n\}_{n=1}^N). \qquad (5)$$

## 3.3 TRAINING UNDER THE GUIDANCE OF LEVEL-SPECIFIC OBJECTIVES

Using only the same discrete intent does not naturally and necessarily inherit the multi-person correlation. Thus, we optimize both the parameters of the predictor and the discrete codes with a level-specific training strategy. We jointly train both levels of individual movement modeling and social interaction modeling, but with each level guided by its own objective. In each forward pass, we explicitly produce different output predictions from different intents of the two levels. And then, in the backward pass, the discrete intent codes $\mathbf{z}^d$ are optimized separately at different levels, while the parameters $\theta$ of the forecasting model $\mathcal{G}$ are updated based on the fused losses from the two levels.

**Local-Level Training.** At the local level, we train the model without social interactions. Given independent multi-person motion data $(\{\mathbf{X}_n\}_{n=1}^N, \{\mathbf{Y}_n\}_{n=1}^N)$, we first randomly sample the discrete intent codes and merge them with independently sampled continuous intent codes into $M \times N$ different latent codes $\{\{\mathbf{z}_n^m\}_{n=1}^N\}_{m=1}^M$. We then use each intent and the past motion of each person to predict $M$ future motion sequences $\{\{\widehat{\mathbf{Y}}_n^m\}_{n=1}^N\}_{m=1}^M$ for each person $\{\mathbf{X}_n\}_{n=1}^N$. Local-level objectives consider single-person fidelity and overall diversity respectively.

**Global-Level Training.** Meanwhile, training is also conducted at the global level to enable the modeling of social interactions. The difference from the local setting is that we incorporate the discrete and continuous codes into only $M$ distinct latent codes $\{\{\mathbf{z}^m\}_{n=1}^N\}_{m=1}^M$; hence, the discrete latent codes $\mathbf{z}_d^m$ of all $N$ individuals are the same for a certain $m$-th prediction. Here, we introduce learning objectives to facilitate multi-person fidelity and accuracy.

Please refer to Sec. C of the Appendix for more detail on the aforementioned level-specific learning objectives. Also, Sec. G of the Appendix further demonstrates that these learning objectives are important, and in some cases critical, to the fidelity and diversity of multi-person motion.

**Inference.** During inference, we only use the global-level strategy by sampling the same intent for all individuals present in the scene. Note that the uncertainty of human motion substantially

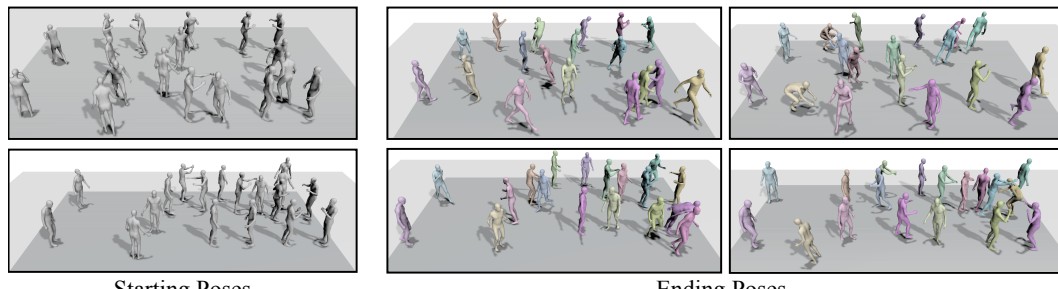

|  | Starting Poses |  | Ending Poses |  |
|---|---|---|---|---|

Figure 3: Qualitative results of DuMMF with a DDPM. We demonstrate the generalizability of our method to handle *a significantly more complex scenario with 18 persons*. Note that our model is trained only on 3-person data. We visualize the predicted final poses at 2 seconds.

Table 1: Quantitative results of DuMMF with a DDPM. Both the baseline and our models are trained using SMPL-X representations on AMASS, and we convert them to skeletons for evaluation. Using the same backbone and generative model, our DuMMF framework significantly provides more accurate predictions with more intents.

| Method, # of Intents | @25 frames | | @50 frames | | @75 frames | |
|---|---|---|---|---|---|---|
| | ADE ↓ | FDE ↓ | ADE ↓ | FDE ↓ | ADE ↓ | FDE ↓ |
| DDPM (Ho et al., 2020), N/A | 2.021 | 4.608 | 3.766 | 7.585 | 5.434 | 11.111 |
| + DuMMF (Ours), 1 | 2.405 | 4.814 | 3.992 | 7.338 | 5.197 | 8.610 |
| + DuMMF (Ours), 3 | **1.456** | **3.316** | **2.700** | **4.823** | **3.513** | **4.880** |

increases with the time horizons of the forecasting. So instead of predicting a fixed number of diverse outputs (Mao et al., 2021; Yuan & Kitani, 2020), in our evaluation, the number grows with the length of the prediction. To this end, we employ autoregressive inference (Wang et al., 2021b) with *progressive diversity*. Given the past motion sequence $\{\mathbf{X}_n\}_{n=1}^N$ and $M$ sampled intent codes, the model outputs $M$ predictions for each person $\{\{\widehat{\mathbf{Y}}_n^m\}_{n=1}^N\}_{m=1}^M$. And then, given a combination of history, the previous predictions $\{\{\mathbf{X}_n, \mathbf{Y}_n^m\}_{n=1}^N\}_{m=1}^M$, and $M$ new intent codes as input, the model outputs $M^2$ predictions $\{\{\widehat{\mathbf{Y}}_n^m\}_{n=1}^N\}_{m=1}^{M^2}$, *etc.*

### 3.4 NETWORK ARCHITECTURE

Our dual-level modeling framework in conjunction with the discrete learnable intent codes is general and, in principle, does not rely on specific network architectures or generative models. To demonstrate this, we combine our framework with various types of deterministic multi-person motion predictors and different generative models, yielding consistent and significant improvements across all baselines (see Sec. 4). As shown in Figure 2, we abstract the encoder of the multi-person motion predictor into two parts: the local part is responsible for encoding single-person motion, while the global part is responsible for encoding multi-person motion and its interactions. A summary of the multi-person predictors used in the paper is given in Table A of the Appendix.

## 4 EXPERIMENTS

**Datasets.** In the main paper, we show the evaluation on two motion capture datasets, CMU-Mocap (CMU) and MuPoTS-3D (Mehta et al., 2018). CMU-Mocap consists of movement sequences with up to two subjects for each scene. It contains 2,235 recordings performed by 144 different subjects, eight of which include double-person motion. We directly adopt these two-person motions for comparisons in two-person scenarios. For skeletal representation, we follow Wang et al. (2021b) for the train/test split and the preprocess to mix single-person and double-person motion together to synthesize a 3-person motion in each scene. Moreover, we construct an even more challenging scenario where we mix up to more people per scene. Please refer to Sec. H of the Appendix for the detail of this generalized setting. For meshes generated from SMPL-X representations (Pavlakos et al., 2019), we extract the multi-person data in CMU-Mocap from AMASS (Mahmood et al., 2019)

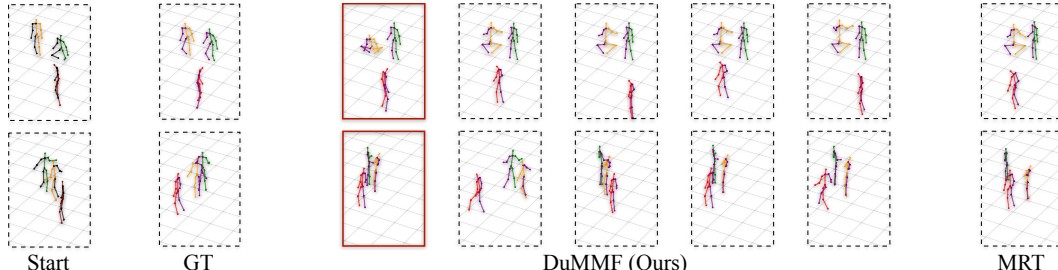

| Start | GT | DuMMF (Ours) | MRT |

Figure 4: Qualitative results of DuMMF with a CGAN model. The leftmost column is the ground truth starting poses, and the second column is the ground truth poses three seconds later. We show our five sampled 3-second predictions in the middle. Our model produces diverse predictions, with one closer to the ground truth (highlighted by the red box) compared with MRT (rightmost column).

Table 2: Quantitative comparison between our DuMMF and deterministic forecasting baselines and their CGAN stochastic variants, using skeletal representations on CMU-Mocap. The number of intents is set to 5 for stochastic forecasting in 3-person (top) and 2-person (bottom) scenarios. DuMMF significantly improves *multi-person* accuracy and diversity *across various architectures and deterministic predictors*. Additionally, our discrete and continuous intent codes are *complementary to each other in most cases*.

| Architecture | Predictor | Diversifier | Variants | @t = 1s | | | @t = 2s | | | @t = 3s | | |
|---|---|---|---|---|---|---|---|---|---|---|---|---|
| | | | | FPD ↑ | ADE ↓ | FDE ↓ | FPD ↑ | ADE ↓ | FDE ↓ | FPD ↑ | ADE ↓ | FDE ↓ |
| RNN | SC-MPF (Adeli et al., 2020) | Deterministic | N/A | N/A | 0.767 | 1.267 | N/A | 1.274 | 2.213 | N/A | 1.779 | 3.262 |
| | | CGAN | | 0.584 | 0.732 | 1.206 | 1.188 | 1.170 | 1.887 | 1.899 | 1.518 | 2.411 |
| | | CGAN+DuMMF (Ours) | w/o Separation | 0.618 | 0.739 | 1.225 | 1.309 | 1.188 | 1.925 | 2.131 | 1.538 | 2.413 |
| | | | w/o Discrete | 0.772 | 0.735 | 1.213 | 1.710 | 1.170 | 1.887 | 3.046 | 1.513 | 2.379 |
| | | | w/o Continuous | **1.051** | 0.811 | 1.368 | **2.090** | 1.339 | 2.226 | **3.326** | 1.791 | 2.966 |
| | | | Full | 1.041 | **0.702** | **1.133** | 2.035 | **1.091** | **1.692** | 3.097 | **1.381** | **2.082** |
| Transformer | SC-MPF (Adeli et al., 2020) | Deterministic | N/A | N/A | 0.697 | 1.167 | N/A | 1.141 | 1.908 | N/A | 1.523 | 2.637 |
| | | CGAN | | 0.454 | 0.681 | 1.111 | 1.036 | 1.059 | 1.649 | 1.743 | 1.346 | 2.094 |
| | | CGAN+DuMMF (Ours) | w/o Separation | 0.624 | 0.676 | 1.099 | 1.461 | 1.038 | 1.592 | 2.527 | 1.305 | 1.954 |
| | | | w/o Discrete | 0.526 | 0.681 | 1.110 | 1.219 | 1.065 | 1.665 | 2.118 | 1.356 | 2.102 |
| | | | w/o Continuous | **0.931** | **0.666** | **1.083** | **2.016** | 1.029 | 1.578 | **3.258** | 1.298 | 1.963 |
| | | | Full | 0.888 | 0.671 | 1.085 | 1.797 | **1.027** | **1.564** | 2.798 | **1.285** | **1.911** |
| | MRT (Wang et al., 2021b) | Deterministic | N/A | N/A | 0.681 | 1.125 | N/A | 1.082 | 1.765 | N/A | 1.427 | 2.438 |
| | | CGAN | | 0.282 | 0.662 | 1.086 | 0.662 | 1.023 | 1.567 | 1.199 | 1.287 | 1.968 |
| | | CGAN+DuMMF (Ours) | w/o Separation | 0.291 | 0.677 | 1.110 | 0.676 | 1.056 | 1.624 | 1.166 | 1.328 | 2.021 |
| | | | w/o Discrete | 0.169 | 0.669 | 1.093 | 0.414 | 1.049 | 1.674 | 0.783 | 1.353 | 2.184 |
| | | | w/o Continuous | 0.403 | 0.673 | 1.094 | 0.894 | 1.035 | 1.584 | 1.526 | 1.306 | 2.004 |
| | | | Full | **0.716** | **0.658** | **1.053** | **1.435** | **0.993** | **1.472** | **2.206** | **1.232** | **1.823** |
| Transformer | MRT (Wang et al., 2021b) | Deterministic | N/A | N/A | 0.685 | 1.138 | N/A | 1.152 | 2.024 | N/A | 1.624 | 3.018 |
| | | CGAN | | 0.675 | 0.688 | 1.442 | 1.391 | 1.094 | 1.750 | 2.205 | 1.443 | 2.398 |
| | | CGAN+DuMMF (Ours) | w/o Separation | 0.753 | 0.663 | 1.069 | 1.499 | 1.013 | 1.533 | 2.259 | 1.284 | 1.970 |
| | | | w/o Discrete | 0.149 | 0.692 | 1.161 | 0.334 | 1.144 | 1.937 | 0.559 | 1.562 | 2.770 |
| | | | w/o Continuous | 0.827 | **0.623** | **1.010** | 1.709 | **0.971** | 1.485 | 2.692 | **1.226** | 1.823 |
| | | | Full | **1.211** | 0.656 | 1.052 | **2.393** | 0.992 | **1.470** | **3.716** | 1.238 | **1.796** |
| | XIA (Guo et al., 2022) | Deterministic | N/A | N/A | 0.679 | 1.136 | N/A | 1.121 | 1.910 | N/A | 1.529 | 2.727 |
| | | CGAN | | 0.578 | 0.667 | 1.088 | 1.208 | 1.044 | 1.631 | 1.882 | 1.340 | 2.111 |
| | | CGAN+DuMMF (Ours) | w/o Separation | 0.604 | 0.657 | 1.068 | 1.183 | 1.018 | 1.589 | 1.746 | 1.311 | 2.071 |
| | | | w/o Discrete | 0.433 | 0.662 | 1.080 | 1.025 | 1.041 | 1.663 | 1.757 | 1.355 | 2.183 |
| | | | w/o Continuous | 0.234 | 0.669 | 1.113 | 0.458 | 1.090 | 1.814 | 0.720 | 1.463 | 2.507 |
| | | | Full | **1.143** | **0.634** | **1.010** | **2.264** | **0.957** | **1.423** | **3.477** | **1.197** | **1.764** |

and follow the same strategy to mix single-person and double-person motion together. MuPoTS-3D consists of more than 8,000 frames with up to three subjects. We convert the data to the same 15-joint human skeleton and length units as CMU-Mocap, and evaluate the generalization on MuPoTS-3D of a model trained only on CMU-Mocap. We also report our performance on the SoMoF benchmark (Adeli et al., 2020; 2021) in Sec. G of the Appendix.

**Metrics.** For evaluating multi-person motion accuracy and diversity, we adopt the common metrics used in stochastic forecasting (Mao et al., 2021; Yuan & Kitani, 2020; 2019; Salzmann et al., 2020; Kothari et al., 2021) as follows. For accuracy measurement, we follow the Best-of-N (BoN) evaluation and use (a) **Average Displacement Error (ADE)**: the average $\ell_2$ distance over time between the ground truth and the prediction *closest* to the ground truth; (b) **Final Displacement Error (FDE)**: the $\ell_2$ distance between the final pose of the ground truth and the last predicted pose *closest* to the ground truth. For diversity measurement, we employ (c) **Final Pairwise Distance (FPD)**: the average $\ell_2$ distance between all predicted final pose pairs. We disentangle the local pose and the global trajectory of the motion and measure their accuracy and diversity separately by defining the following metrics: **rootADE**, **rootFDE**, **poseADE**, **poseFDE**, **rootFPD**, and **poseFPD**. To measure three different aspects including single-person fidelity, multi-person fidelity, and overall diversity comprehensively, we provide a summary and analysis of all metrics for this novel task in Table F and Sec. F of the Appendix. We include results on all above metrics in Sec. G of the Appendix.

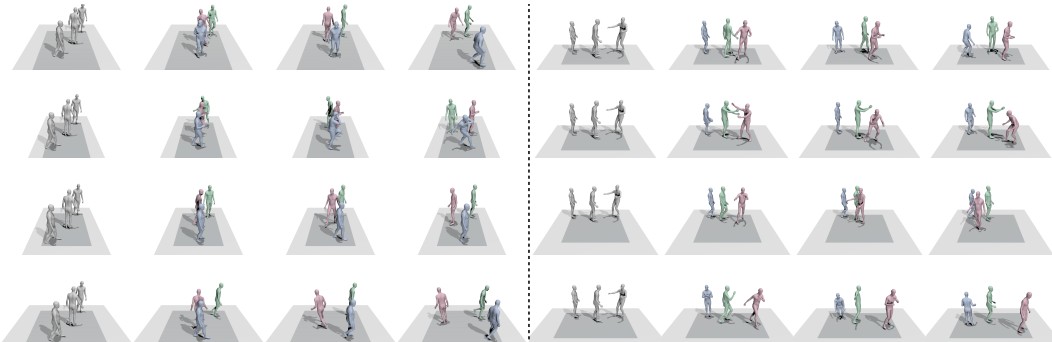

Figure 5: Qualitative results of DuMMF with a DDPM model. We visualize the predicted frames for two different 3-person input motions, which are listed on the left and right respectively. For each input, we generate four sampled motions, arranged in a single column and listed sequentially in time at 0, 1, 2, and 3 seconds. Our method effectively produces diverse human-like social interactions.

**Baselines.** Given that we propose a new task, there is no readily available baseline from existing work. For comparison purposes, we customize several baselines from deterministic multi-person motion forecasting which we summarize in Table A, and extend them to stochastic forecasting. (a) **RNN+SC-MPF** (Adeli et al., 2020): We use Social Pooling (Alahi et al., 2016) to integrate the multi-person information. We follow the implementation in Adeli et al. (2020) and reproduce the results. For the Social Pooling module, we select the maximum pooling for its best performance as ablated in Adeli et al. (2020). (b) **Transformer+SC-MPF**: For a fair comparison, we introduce a transformer-based encoder architecture to combine with the Social Pooling module. (c) **XIA** (Guo et al., 2022): We follow their implementation and use Cross Interaction Attention (XIA) to encode 2-person information. (d) **MRT** (Wang et al., 2021b): We use multi-range transformer (MRT) as our main architecture for its state-of-the-art performance. We further extend MRT to 2-person scenarios to compare with **XIA**. For stochastic forecasting, we formulate a conditional generative adversarial network (CGAN) (Mirza & Osindero, 2014) and a denoising diffusion probabilistic model (DDPM) (Ho et al., 2020), with or without dual-level modeling and discrete intents.

**Implementation Details.** For the skeletal representation, following Wang et al. (2021b), we train the model to predict a 15-frame sequence of 3 people given the ground truth 15, 30, and 45 past frames at 15Hz, as the encoder (RNN/Transformer) accepts different input lengths. We use a 6-layer transformer (or RNNs), where we set the feature dimension to 128. For evaluation, we recursively predict the next 15 frames 3 times given all past frames generated, as illustrated in Sec. 3.3. Thus, given the number of intents to be $M$, the model outputs $M$, $M^2$, and $M^3$ different predictions in sequence. For the SMPL representation, we train the model to predict a 25-frame sequence of 3 people given the 10 past frames at 30Hz. We use an 8-layer transformer, where we set the feature dimension to 512. For evaluation, similarly, we use 10 frames as the past motion and recursively predict the next 25 frames. Additional implementation details are provided in Sec. E of the Appendix.

**Quantitative Results.** We compare our method with a pure DDPM in Table 1. Our improvement is significant. Notably, even in the case of a single intent, where we only evaluate one prediction, our method outperforms DDPM in long-term generation. As the number of intents increases, our method provides more accurate results, especially in long-term prediction. In Table 2, we demonstrate that our DuMMF framework benefits all predictor variants. We observe that the simple combination of deterministic predictors and CGAN results in very low diversity and accuracy. By contrast, our full approach significantly outperforms CGAN baselines on both diversity and accuracy across all backbones. In Table D of the Appendix, we further show that DuMMF achieves the best generalization results across all predictors on MuPoTS-3D, highlighting its *generality* and *superiority*.

**Ablation: Effectiveness of Dual-Level Modeling.** Tables 2 and 3 show the *effectiveness* of our dual-level framework. First, we investigate the settings with only single-person motion modeling or only social interaction modeling. In Table 3, compared with our full method, modeling independent multi-person motion ('w/o Social') provides higher diversity but leads to inaccurate poses, since the social restrictions are not considered. With only social interaction modeling, the model ('w/o Individual') cannot output sufficiently diverse predictions, which also makes predictions inaccurate,

Table 3: Ablation study of our DuMMF with a CGAN and MRT (Wang et al., 2021b) using skeletal representations on CMU-Mocap and MuPoTS-3D. We report both accuracy and diversity for root and pose separately. The results show the effectiveness of our dual-level modeling along with discrete motion intents, and *complementariness* of local-level and global-level modeling.

| Method @$t = 3s$ | CMU-Mocap (CMU) | | | | | | Generalized to MuPoTS-3D (Mehta et al., 2018) | | | | | |
| | ROOT | | | POSE | | | ROOT | | | POSE | | |
| | FPD ↑ | ADE ↓ | FDE ↓ | FPD ↑ | ADE ↓ | FDE ↓ | FPD ↑ | ADE ↓ | FDE ↓ | FPD ↑ | ADE ↓ | FDE ↓ |
|---|---|---|---|---|---|---|---|---|---|---|---|---|
| MRT (Wang et al., 2021b) | N/A | 0.753 | 0.946 | N/A | 0.301 | 0.567 | N/A | 0.603 | 0.835 | N/A | 0.220 | 0.461 |
| Ours, Deterministic | N/A | **0.738** | **0.916** | N/A | **0.288** | **0.526** | N/A | **0.587** | **0.809** | N/A | **0.203** | **0.411** |
| w/o Individual | 0.178 | **0.733** | **0.896** | 0.339 | 0.264 | 0.453 | 0.120 | **0.584** | 0.797 | 0.253 | 0.180 | 0.331 |
| w/o Social | **1.661** | 0.751 | 0.909 | **1.077** | 0.264 | 0.413 | **2.100** | 0.605 | 0.818 | **1.283** | 0.183 | 0.318 |
| w/o Separation | 0.167 | 0.747 | 0.911 | 0.293 | 0.271 | 0.448 | 0.105 | 0.596 | 0.815 | 0.213 | 0.202 | 0.386 |
| Full | 0.249 | 0.734 | 0.898 | 0.562 | **0.243** | **0.390** | 0.187 | 0.588 | 0.796 | 0.444 | **0.170** | **0.305** |
| w/o Discrete | 0.114 | 0.761 | 0.949 | 0.196 | 0.279 | 0.490 | 0.083 | 0.620 | 0.857 | 0.158 | 0.212 | 0.411 |
| w/o Continuous | 0.208 | 0.746 | 0.905 | 0.381 | 0.265 | 0.439 | 0.159 | 0.594 | 0.801 | 0.328 | 0.196 | 0.378 |
| Full | **0.249** | **0.734** | **0.898** | **0.562** | **0.243** | **0.390** | **0.187** | **0.588** | **0.796** | **0.444** | **0.170** | **0.305** |

as more varied outputs have a better chance of covering the ground truth. Note that the results of 'CGAN' and 'w/o Separation' in Table 2 and 'w/o Separation' in Table 3 are worse since they simply use all the learning objectives together without disentangling the two levels of modeling, while 'w/o Separation' is slightly better due to the use of discrete intents. Under our dual-level framework with level-specific motion intents and learning objectives, the model can more effectively incorporate the benefits of both levels, thus leading to improved accuracy and diversity. In Sec. G of the Appendix, we further demonstrate that our dual-level benefits different predictor variants.

**Ablation: Effectiveness of Discrete Human Intents.** In Tables 2 and 3, we also demonstrate that discrete human intents are effective and crucial. We observe the best results when using both discrete and continuous intents, indicating that they are complementary. In the absence of discrete intents ('w/o Discrete'), the performance is only comparable with the baseline ('CGAN'). Importantly, with the help of discrete intents, the improvement of dual-level modeling ('Full') over 'w/o Separation' is more pronounced, compared with the improvement of 'w/o Discrete' over 'CGAN.' Therefore, discrete learnable intents are essential for effectively integrating the advantages of both levels during training. The performance without continuous intents ('w/o Continuous') is slightly worse than the full method. Our hypothesis is that relying solely on discrete intents is limiting, because they only support a finite number of outputs. In Sec. G of the Appendix, we further investigate how the number of discrete intents impacts stochastic forecasting.

**Qualitative Results.** Consistent with the quantitative evaluation above, we observe that our method provides diverse multi-person motion, and produces predictions closer to the ground truth compared with the deterministic method MRT in Figure 4. In Figure 5, we qualitatively show that our generated results in meshes reflect real-world diversity of social interactions. Furthermore, we provide qualitative results for more-person scenarios in Figure 3, and others in Figure A and Figure B of the Appendix. Please refer to Sec. H of the Appendix for more detail on the more-person setting.

**Limitation.** Although our dual-level framework has proven effective in producing high-quality and diverse predictions, we have observed artifacts such as foot skating in some predicted motion sequences. This is because our model relies solely on loss functions to constrain motion, rather than explicitly modeling articulated motion. As this is a common issue in learning-based methods, we plan to exploit a physical simulator to further improve the plausibility of our predicted motion.

## 5   CONCLUSION

We formulate a novel task called stochastic multi-person 3D motion forecasting, which better reflects the real-world human motion complexities. To simultaneously achieve single-person fidelity, social realism, and overall diversity, we propose a dual-level generative modeling framework (DuMMF) with learnable latent intent codes. Compared with prior work on deterministic or single-person prediction, our model learns to generate diverse and realistic human motion and interactions. Notably, our framework is model-agnostic and generalizes to unseen more-person scenarios.

**Ethics Statement.** Our proposed technique is useful in many applications, such as self-driving to avoid crowds. The potential negative societal impacts include: (a) our approach can be used to synthesize highly realistic human motion, which might lead to the spread of false information; (b) our approach requires real behavioral information as input, which may raise privacy concerns and result in the disclosure of sensitive identity information. Nevertheless, our model operates on the processed human skeleton representation that contains minimal identifying information, unlike raw data. On the positive side, this can be seen as a privacy-enhancing feature.

**Acknowledgement.** This work was supported in part by NSF Grant 2106825, NIFA Award 2020-67021-32799, the Jump ARCHES endowment through the Health Care Engineering Systems Center, the National Center for Supercomputing Applications (NCSA) at the University of Illinois at Urbana-Champaign through the NCSA Fellows program, the IBM-Illinois Discovery Accelerator Institute, the Illinois-Insper Partnership, and the Amazon Research Award. This work used NVIDIA GPUs at NCSA Delta through allocation CIS220014 from the Advanced Cyberinfrastructure Coordination Ecosystem: Services & Support (ACCESS) program, which is supported by NSF Grants #2138259, #2138286, #2138307, #2137603, and #2138296.

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

In this Appendix, we include additional method details and experimental results that are not included in the main paper due to limited space as follows. 1) We provide a visualization video as additional qualitative results, and the details are explained in Sec. A. 2) We include a further discussion on related work in Sec. B. 3) We explain different generative models incorporated in our proposed dual-level modeling framework in Sec. D; 4) We provide additional details of the experimental implementation in Sec. E and the summary of evaluation metrics in Sec. F. 5) To elaborate the effectiveness of our method, we provide additional ablation experiments with qualitative and quantitative analysis in Sec. G, and we evaluate our approach in more challenging scenarios with a significantly increased number of people in Sec. H.

## A    VISUALIZATION VIDEO

In addition to Figure 4 and Figure 5 in the main paper and more qualitative results in this Appendix (Figures A and B), we provide a video to demonstrate more comprehensive visualizations of multi-person 3D motion forecasting at `https://sirui-xu.github.io/DuMMF/images/demo.mp4`. In this video, we illustrate that our method DuMMF generates diverse multi-person motion and social interactions, as well as taking into account both single-person and multi-person fidelity. We also show that our model is *scalable* and provide effective predictions in more challenging scenarios with a significantly increased number of people and associated more complex interactions. We also highlight the impact and effectiveness of our dual-level modeling framework.

## B    ADDITIONAL DISCUSSION ON RELATED WORK

As we demonstrate in the main paper, our proposed stochastic multi-person motion forecasting needs to *simultaneously* take into account single-person pose fidelity, consistency of pose and trajectory, social interactions between poses, and overall diversity of motion, while prior work including stochastic multi-person trajectory forecasting (Alahi et al., 2014; Kothari et al., 2020; 2021) and deterministic multi-person motion forecasting (Adeli et al., 2020; 2021; Guo et al., 2022; Wang et al., 2021b) focuses on simplified scenarios. A survey (Barquero et al., 2022) summarizes the progress in this area. Our problem reveals real-world complexity with substantially increased, multi-faced challenges which have not been, and cannot be, jointly tackled by any of the prior literature. In particular, in our stochastic scenario, the fidelity and interactions need to be both satisfied and diversified for all the predictions, which is very challenging and cannot be addressed by a simple extension of existing work on stochastic multi-person trajectory forecasting and deterministic multi-person motion forecasting as shown in Sec. G.

For **deterministic multi-person motion forecasting**, in Sec. 2 of the main paper, we have highlighted our key difference with Adeli et al. (2020; 2021); Guo et al. (2022), and here we discuss in more detail. Adeli et al. (2020) introduce a social pooling module to integrate social information. Adeli et al. (2021) propose a graph attentional network to jointly model human-human and human-object interactions. Both methods utilize additional contextual information to aid deterministic prediction with their proposed Social Motion Forecasting (SoMoF) benchmark (Adeli et al., 2020; 2021). Guo et al. (2022) propose a cross-interaction attention mechanism to predict cross dependencies between two pose sequences, making it applicable only to 2-person scenarios.

There are many methods achieving strong performance with simple and non-transformer-based frameworks on the SoMoF leaderboard. Specifically, futuremotion_ICCV21 (Wang et al., 2021a) develops a simple but effective framework based on a combination of graph convolutional networks and multiple motion optimization techniques. DViTA (Parsaeifard et al., 2021) decouples the human pose into a trajectory and local pose and employs a VAE to learn a representation of the local pose dynamics. LSTMV_LAST (Saadat et al., 2021) proposes a sequence-to-sequence LSTM via keypoint-based representation.

Complementing Sec. 2 of the main paper, we here discuss in more depth the **trajectory-level multi-person forecasting**. Trajectron++ (Salzmann et al., 2020) constructs a spatio-temporal scene graph where nodes represent individuals and edges represent their interactions, and global and local modeling is embedded in this graph structure. EvolveGraph (Li et al., 2020) introduces an effective dynamic mechanism to evolve interaction graphs, which can flexibly model dynamically changing interactions. Tra2tra (Xu et al., 2021) introduces a global social spatio-temporal attention neural

Table A: Summary of methods for encoding and integrating multi-person motions and interactions. We follow to abstract the encoder into local and global part as Figure 2(d). As we use the same decoder for baselines (Sec. 4 of the main paper), we only discuss encoders of three predictors in this paper. Note that XIA (Guo et al., 2022) can only be applied to two-person scenarios.

| Method | Local Encoder | Global Encoder |
|---|---|---|
| **SC-MPF (Adeli et al., 2020)** | GRU($\mathbf{X_i}$) | MaxPool($\{$GRU($\mathbf{X_n}$)$\}_{n=1}^N$) |
| **MRT (Wang et al., 2021b)** | MultiHead($\mathbf{X_i}, \mathbf{X_i}, \mathbf{X_i}$) | MultiHead($\{\mathbf{X_n}\}_{n=1}^N, \{\mathbf{X_n}\}_{n=1}^N, \{\mathbf{X_n}\}_{n=1}^N$) |
| **XIA (Guo et al., 2022)** | N/A | $\{$MultiHead($\mathbf{X_1}, \mathbf{X_2}, \mathbf{X_2}$, MultiHead($\mathbf{X_2}, \mathbf{X_1}, \mathbf{X_1}$)$\}$ |

network to encode both spatial interactions and temporal features. GroupNet (Xu et al., 2022a) employs a multi-scale hypergraph neural network that models group-based interactions and facilitates more comprehensive relational reasoning. T-GNN (Xu et al., 2022c) introduces a transferable graph neural network that allows not only trajectory prediction but also domain alignment of potential distribution differences. MID (Gu et al., 2022) employs a diffusion model to model the variation of indeterminacy for trajectory prediction.

# C  ADDITIONAL DETAILS OF LEVEL-SPECIFIC OBJECTIVES

We use $\Delta$ to represent the residual of the motion sequence. For example, $\Delta \widehat{\mathbf{Y}}_i^j = [\widehat{\mathbf{Y}}_i^j[T_h + 1] - \mathbf{X}_i[T_h], \widehat{\mathbf{Y}}_i^j[T_h + 2] - \widehat{\mathbf{Y}}_i^j[T_h + 1], \ldots, \widehat{\mathbf{Y}}_i^j[T_h + T_p] - \widehat{\mathbf{Y}}_i^j[T_h + T_p - 1]]$.

**Local-Level Objectives.** We adopt the multiple output loss (Guzmán-rivera et al., 2012) and extend it to the local reconstruction loss of multiple people $\mathcal{L}_{\mathrm{lR}}$, which is used to optimize the most accurate prediction of each person while maintaining diversity. We highlight the structure of the human skeleton by introducing the limb loss (Mao et al., 2021) $\mathcal{L}_{\mathrm{L}}$. Specifically,

$$\mathcal{L}_{\mathrm{lR}} = \frac{1}{N} \sum_{n=1}^N \min_{m=1,\ldots,M} \|\Delta\widehat{\mathbf{Y}}_n^m - \Delta\mathbf{Y}_n\|_2^2, \tag{6}$$

$$\mathcal{L}_{\mathrm{L}} = \frac{1}{N * M} \sum_{n=1}^N \sum_{m=1}^M \|\widehat{\mathbf{L}}_n^m - \mathbf{L}_n\|_2^2, \tag{7}$$

where the vector $\mathbf{L}_n$ represents the ground truth distance between all pairs of joints that are physically connected in the $n$-th human body and $\widehat{\mathbf{L}}_n^m$ includes the limb length for all the $T_p$ poses in $\widehat{\mathbf{Y}}_n^m$.

We further develop a multimodal reconstruction loss $\mathcal{L}_{\mathrm{mmR}}$ to provide additional supervision for all outputs $\{\{\Delta\widehat{\mathbf{Y}}_n^m\}_{n=1}^N\}_{m=1}^M$. We first construct pseudo future motion $\{\widetilde{\mathbf{Y}}_i^p\}_{p=1}^P$ for each historical sequence $\mathbf{X}_i$. Different from (Yuan & Kitani, 2020; Mao et al., 2021), we additionally consider translation $\mathbf{T} \in \mathbb{R}^3$ and rotation $\mathbf{R} \in \mathbb{R}^{3\times3}$ of the pose. Specifically, given a threshold $\epsilon$, we cluster future motion with a similar start pose and train the model with their residuals as

$$\{\widetilde{\mathbf{Y}}_i^p\}_{p=1}^P = \{\widetilde{\mathbf{Y}}_i^p | \min_{\mathbf{R}, \mathbf{T}} \|\mathbf{R}(\widetilde{\mathbf{X}}_i^p[T_h] - \mathbf{T}) - \mathbf{X}_i[T_h]\|_2 \leq \epsilon\}, \tag{8}$$

$$\mathcal{L}_{\mathrm{mmR}} = \frac{1}{N * P} \sum_{n=1}^N \sum_{p=1}^P \min_{m=1,\ldots,M} \|\Delta\widehat{\mathbf{Y}}_n^m - \Delta\mathbf{Y}_n^p\|_2^2. \tag{9}$$

To explicitly encourage diversity, we adopt a diversity-promoting loss (Yuan & Kitani, 2020), which directly promotes the pairwise distance between the predictions of a single person. We decompose this loss into two parts, promoting the diversity of local pose and the global root separately. Supposing that $\widehat{\mathbf{Y}}_n^m(l)$ and $\widehat{\mathbf{Y}}_n^m(g)$ are the local pose and the global root joint extracted from the global pose $\widehat{\mathbf{Y}}_n^m$, respectively, and $\alpha$ and $\beta$ are two hyperparameters, this diversity-promoting loss is denoted as

$$\mathcal{L}_{\mathrm{D}} = \frac{1}{N * M(M-1)} \sum_{n=1}^N \sum_{m=1}^M \sum_{k=m+1}^M [\exp(\frac{\|\widehat{\mathbf{Y}}_n^m(g) - \widehat{\mathbf{Y}}_n^k(g)\|_2^2}{\alpha}) + \exp(\frac{\|\widehat{\mathbf{Y}}_n^m(l) - \widehat{\mathbf{Y}}_n^k(l)\|_2^2}{\beta})]. \tag{10}$$

Table B: Summary of the *complementary* evaluation metrics in the multi-person 3D motion forecasting task, with each focusing on evaluating different aspects of predicted motion. Here for simplicity, we show the metrics *without alignment*. We also provide ADE, FDE, and FPD with alignment to evaluate the pose and trajectory separately, as explained in Sec. 4 of the main paper.

| Type | Metrics | Definition |
|---|---|---|
| **Single-Person Fidelity** | Local Average Displacement Error (lADE) | $\frac{1}{NT_p} \sum_{n=1}^{N} \min_m \|\widehat{\mathbf{Y}}_n^m - \mathbf{Y}_n\|_2$ |
| | Local Final Displacement Error (lFDE) | $\frac{1}{N} \sum_{n=1}^{N} \min_m \|\widehat{\mathbf{Y}}_n^m[T_p] - \mathbf{Y}_n[T_p]\|_2$ |
| | Foot Skating Ratio (FSR) | average ratio of frames where both foot joints are close to the ground ($\leq$ 5cm) and fast ($\geq$ 75mm/s) |
| **Multi-Person Fidelity** | (Global) Average Displacement Error (ADE) | $\min_m \frac{1}{NT_p} \sum_{n=1}^{N} \|\widehat{\mathbf{Y}}_n^m - \mathbf{Y}_n\|_2$ |
| | (Global) Final Displacement Error (FDE) | $\min_m \frac{1}{N} \sum_{n=1}^{N} \|\widehat{\mathbf{Y}}_n^m[T_p] - \mathbf{Y}_n[T_p]\|_2$ |
| | Trajectory Collision Ratio (TCR) | average ratio of frames where there is a collision between any two trajectories |
| | Average Human Displacement (AHD) | $\frac{1}{NM} \sum_{n=1}^{N} \sum_{m=1}^{M} \|\widehat{\mathbf{Y}}_n^m[T_p] - \widehat{\mathbf{Y}}_n^m[1]\|_2$ |
| **Overall Diversity** | Final Pairwise Distance (FPD) | $\frac{1}{NM(M-1)} \sum_{n=1}^{N} \sum_{m=1}^{M} \sum_{k=m+1}^{M} \|\widehat{\mathbf{Y}}_n^m[T_p] - \widehat{\mathbf{Y}}_n^k[T_p]\|_2$ |

For CGAN, a GAN loss (Kocabas et al., 2020) is leveraged to train the model and the local discriminator $\mathcal{D}_l$ for individual body realism. Suppose $\{\mathbf{Y}_n^*\}_{n=1}^N$ is the set of real motion clips sampled from the data, we have the following.

$$\mathcal{L}_{\text{lGAN}} = \frac{1}{N*M} \sum_{n=1}^{N} \sum_{m=1}^{M} \|\mathcal{D}_l(\widehat{\mathbf{Y}}_n^m)\|_2^2 + \frac{1}{N} \sum_{n=1}^{N} \|\mathcal{D}_l(\mathbf{Y}_n^*) - \mathbf{1}\|_2^2. \quad (11)$$

**Global-Level Objectives.** Since in this setting we treat $N$ individuals as a whole, the reconstruction loss is reformulated as

$$\mathcal{L}_{\text{gR}} = \min_{m=1,\dots,M} \frac{1}{N} \sum_{n=1}^{N} \|\Delta\widehat{\mathbf{Y}}_n^m - \Delta\mathbf{Y}_n\|_2^2. \quad (12)$$

For CGAN, a global GAN loss is further leveraged to promote the realism of social motion, where the global discriminator $\mathcal{D}_g$ takes the motion of all $N$ people as input. Suppose $\{\mathbf{Y}_n^{**}\}_{n=1}^N$ is the multi-person motion clip sampled from the data, and we have

$$\mathcal{L}_{\text{gGAN}} = \frac{1}{M} \sum_{m=1}^{M} \|\mathcal{D}_g(\{\widehat{\mathbf{Y}}_n^m\}_{n=1}^N)\|_2^2 + \|\mathcal{D}_g(\{\mathbf{Y}_n^{**}\}_{n=1}^N) - \mathbf{1}\|_2^2. \quad (13)$$

Table C: Quantitative results (w/ error bar) of our DuMMF on *single-person* accuracy.

| Method, # of Intents | @t = 1s | | @t = 2s | | @t = 3s | |
|---|---|---|---|---|---|---|
| | lADE ↓ | lFDE ↓ | lADE ↓ | lFDE ↓ | lADE ↓ | lFDE ↓ |
| DuMMF (Ours), 2 | $0.663 \pm 0.007$ | $1.088 \pm 0.019$ | $1.029 \pm 0.019$ | $1.612 \pm 0.043$ | $1.308 \pm 0.031$ | $2.014 \pm 0.086$ |
| DuMMF (Ours), 3 | $0.656 \pm 0.007$ | $1.056 \pm 0.007$ | $0.996 \pm 0.009$ | $1.503 \pm 0.032$ | $1.244 \pm 0.021$ | $1.883 \pm 0.076$ |
| DuMMF (Ours), 5 | $0.648 \pm 0.011$ | $1.033 \pm 0.022$ | $0.963 \pm 0.020$ | $1.400 \pm 0.035$ | $1.176 \pm 0.025$ | $1.672 \pm 0.052$ |

# D ADDITIONAL DETAILS OF GENERATIVE MODELS

In this paper, we mainly evaluate the generalizability of our method by incorporating it with a CGAN (Mirza & Osindero, 2014) or a DDPM (Ho et al., 2020) model. For CGAN, we introduce local discriminator and global discriminator for objectives at local and global levels, respectively. For the model incorporated with DDPM, we disregard these two GAN losses.

**Local Discriminator.** To ensure the fidelity of the motion, especially to address the problem of *foot skating*, where the feet appear to slide in the ground plane, we concatenate the predicted motion of each person $\mathbf{Y}_i^j$ with the feet velocities $\Delta \mathbf{F}_i^j$ as input to a local discriminator $\mathcal{D}_l$. The local discriminator uses a local-range transformer encoder adopted from Wang et al. (2021b).

**Global Discriminator.** To ensure the realism of social interactions and avoid motion collisions, we propose a global discriminator $\mathcal{D}_g$ that encodes all motion of $N$ people $\{\mathbf{Y}_n^j\}_{n=1}^N$ at the same time and outputs a fidelity score. The global discriminator uses a global-range transformer encoder adopted from Wang et al. (2021b).

## E ADDITIONAL IMPLEMENTATION DETAILS

Here, we provide more details on the implementation of our method DuMMF. The two hyperparameters $(\alpha, \beta)$ in diversity promoting loss (Sec. C) are set to $(50, 100)$. The model is trained using a batch size of 32 for 50 epoch, with 6000 training examples per epoch. We use ADAM (Kingma & Ba, 2014) to train the model. The code is based on PyTorch. On one NVIDIA GeForce GTX TITAN X GPU, training an epoch takes approximately 5 minutes. For license, CMU-Mocap (CMU) is free for all users; MuPoTS-3D (Mehta et al., 2018) is for noncommercial purposes. Part of our code is based on AMCParser (MIT license), attention-is-all-you-need-pytorch (MIT license), and MRT (Wang et al., 2021b) (not specified), and XIA (Guo et al., 2022) (GPL license).

Table D: Quantitative comparison on MuPoTS-3D between our DuMMF and deterministic forecasting baselines and their CGAN variants. All models are trained only using skeletal representations on CMU-Mocap and we compare their generalizations on MuPoTS-3D here. The number of intents is set to 5 for stochastic forecasting on 3-person (top) and 2-person (bottom) scenarios. DuMMF significantly improves *multi-person* accuracy and diversity across various architectures and deterministic predictors.

| Architecture | Predictor | Diversifier | Variants | @t=1s | | | @t=2s | | | @t=3s | | |
|---|---|---|---|---|---|---|---|---|---|---|---|---|
| | | | | FPD↑ | ADE↓ | FDE↓ | FPD↑ | ADE↓ | FDE↓ | FPD↑ | ADE↓ | FDE↓ |
| RNN | SC-MPF (Adeli et al., 2020) | Deterministic | N/A | N/A | 0.514 | 0.880 | N/A | 0.902 | 1.632 | N/A | 1.319 | 2.594 |
| | | CGAN | N/A | 0.467 | 0.473 | 0.786 | 0.846 | 0.753 | 1.207 | 1.215 | 0.967 | 1.518 |
| | | CGAN+DuMMF (Ours) | w/o Separation | 0.511 | 0.481 | 0.801 | 0.987 | 0.772 | 1.252 | 1.471 | 1.001 | 1.600 |
| | | | w/o Discrete | 0.697 | 0.478 | 0.793 | 1.534 | 0.770 | 1.258 | 2.800 | 1.008 | 1.618 |
| | | | w/o Continuous | 0.736 | 0.627 | 1.092 | 1.412 | 1.074 | 1.860 | 2.262 | 1.476 | 2.588 |
| | | | Full | **0.747** | **0.455** | **0.744** | **1.441** | **0.707** | **1.103** | **2.200** | **0.893** | **1.364** |
| Transformer | SC-MPF (Adeli et al., 2020) | Deterministic | N/A | N/A | 0.430 | 0.731 | N/A | 0.723 | 1.258 | N/A | 0.995 | 1.781 |
| | | CGAN | N/A | 0.504 | 0.403 | **0.690** | 1.099 | 0.659 | 1.077 | 1.834 | 0.857 | 1.371 |
| | | CGAN+DuMMF (Ours) | w/o Separation | 0.787 | 0.407 | 0.701 | 1.791 | 0.667 | 1.080 | 3.047 | 0.865 | 1.363 |
| | | | w/o Discrete | 0.557 | 0.403 | 0.693 | 1.202 | 0.657 | 1.075 | 1.975 | 0.857 | 1.382 |
| | | | w/o Continuous | **0.917** | 0.406 | 0.707 | **2.124** | 0.685 | 1.163 | **3.535** | 0.924 | 1.573 |
| | | | Full | 0.865 | **0.402** | 0.691 | 1.816 | **0.656** | **1.071** | 2.902 | **0.852** | **1.352** |
| | MRT (Wang et al., 2021b) | Deterministic | N/A | N/A | 0.427 | 0.747 | N/A | 0.747 | 1.354 | N/A | 1.071 | 2.043 |
| | | CGAN | N/A | 0.176 | 0.421 | 0.741 | 0.452 | 0.729 | 1.288 | 0.878 | 1.017 | 1.846 |
| | | CGAN+DuMMF (Ours) | w/o Separation | 0.194 | 0.423 | 0.743 | 0.456 | 0.731 | 1.267 | 0.855 | 1.000 | 1.754 |
| | | | w/o Discrete | 0.133 | 0.421 | 0.743 | 0.335 | 0.732 | 1.286 | 0.627 | 1.012 | 1.805 |
| | | | w/o Continuous | 0.308 | 0.418 | 0.730 | 0.732 | 0.716 | 1.242 | 1.304 | 0.982 | 1.734 |
| | | | Full | **0.510** | **0.408** | **0.711** | **1.075** | **0.683** | **1.135** | **1.740** | **0.905** | **1.491** |
| Transformer | MRT (Wang et al., 2021b) | Deterministic | N/A | N/A | 0.475 | 0.822 | N/A | 0.853 | 1.605 | N/A | 1.268 | 2.539 |
| | | CGAN | N/A | 0.556 | 0.457 | 0.791 | 1.123 | 0.770 | 1.300 | 1.778 | 1.041 | 1.784 |
| | | CGAN+DuMMF (Ours) | w/o Separation | 0.669 | 0.441 | 0.752 | 1.292 | 0.713 | 1.146 | 1.939 | 0.922 | 1.469 |
| | | | w/o Discrete | 0.100 | 0.475 | 0.846 | 0.231 | 0.843 | 1.523 | 0.402 | 1.192 | 2.207 |
| | | | w/o Continuous | 0.691 | **0.413** | **0.710** | 1.462 | **0.674** | 1.109 | 2.338 | 0.885 | 1.433 |
| | | | Full | **1.133** | 0.432 | 0.734 | **2.283** | 0.694 | **1.104** | **3.575** | **0.891** | **1.384** |
| Transformer | XIA (Guo et al., 2022) | Deterministic | N/A | N/A | 0.478 | 0.840 | N/A | 0.848 | 1.555 | N/A | 1.223 | 2.338 |
| | | CGAN | N/A | 0.489 | 0.434 | 0.743 | 1.022 | 0.706 | 1.145 | 1.611 | 0.918 | 1.474 |
| | | CGAN+DuMMF (Ours) | w/o Separation | 0.479 | 0.435 | 0.741 | 0.902 | 0.705 | 1.140 | 1.310 | 0.913 | 1.464 |
| | | | w/o Discrete | 0.451 | **0.428** | 0.742 | 1.065 | 0.710 | 1.192 | 1.804 | 0.947 | 1.586 |
| | | | w/o Continuous | 0.215 | 0.447 | 0.784 | 0.440 | 0.768 | 1.335 | 0.693 | 1.052 | 1.853 |
| | | | Full | **0.895** | 0.431 | **0.731** | **1.902** | **0.690** | **1.092** | **3.057** | **0.879** | **1.333** |

## F SUMMARY OF EVALUATION METRICS

In the main paper, we mainly focused on evaluating the fidelity and diversity of the predicted multi-person motion based on **Average Displacement Error (ADE)**, **Final Displacement Error (FDE)**, and **Final Pairwise Distance (FPD)**, and briefly discussed other metrics. Here, we explain these and additional metrics in detail and also provide a systematic summary in Table B for better understanding. Additional comparisons based on these metrics are shown in the following sections.

As summarized in Table B, we group the metrics into thee types, with each type evaluating different aspects of predicted motion (discussed in Sec. 4 of the main paper) – **single-person fidelity**, **multi-person fidelity**, and **overall diversity**.

Table E: Quantitative comparison on SoMoF Benchmark. Here, we only show the deterministic forecasting results. Our method with MRT predictor significantly outperform two deterministic baselines. We use VIM (Adeli et al., 2020) as the metric. * means we directly report the results from the benchmark leaderboard.

| Method, # of Intents | Prediction Time | | | | |
|---|---|---|---|---|---|
| | 100 ms | 240 ms | 500 ms | 640 ms | 900 ms |
| SC-MPF* (Adeli et al., 2020), N/A | 46.28 | 73.88 | 130.23 | 160.83 | 208.44 |
| TRiPOD* (Adeli et al., 2021), N/A | 30.26 | 51.84 | 85.08 | 104.78 | 146.33 |
| MRT (Wang et al., 2021b), N/A | **22.93** | **42.35** | 79.41 | 99.02 | 137.93 |

Table F: Quantitative comparisons between our DuMMF and deterministic forecasting baselines and their CGAN variants on SoMoF benchmark. The number of intents is set to 5 for stochastic forecasting. Our DuMMF significantly improves *multi-person* accuracy and diversity.

| Architecture | Predictor | Diversifier | Variants | $@t=1s$ | | |
|---|---|---|---|---|---|---|
| | | | | FPD↑ | ADE↓ | FDE↓ |
| | | Deterministic | N/A | N/A | 0.575 | 0.961 |
| | | CGAN | | 0.501 | 0.582 | 0.983 |
| Transformer | MRT (Wang et al., 2021b) | | w/o Separation | 0.864 | 0.590 | 0.969 |
| | | **CGAN+DuMMF (Ours)** | w/o Discrete | **1.075** | 0.567 | 0.957 |
| | | | w/o Continuous | 1.063 | 0.587 | 1.018 |
| | | | Full | 0.969 | **0.564** | **0.935** |

Specifically, **Local Average Displacement Error (lADE)** and **Local Final Displacement Error (lFDE)** are proposed to further evaluate the *single*-person fidelity. They compute the average distance between the *individual* ground truth and the *individual* prediction *closest* to the *individual* ground truth. Note that ADE (or lADE) and FDE (or lFDE) only measure the best predictions in all outputs. While we can *average* ADE (or lADE) and FDE (or lFDE) by computing the distance between multiple predictions and a single ground truth, this way of assessing the overall prediction quality cannot reflect motion realism. For example, a very realistic but diverse set of outputs may have poor *average* ADE and FDE. Therefore, we do not use such metrics for our evaluation.

For diversity evaluation, a common metric is Average Pairwise Distance (APD) (Mao et al., 2021; Yuan & Kitani, 2020; 2019) that is the average $\ell_2$ distance between all predicted *motion* pairs. In this paper, we formulate diverse forecasting as producing more forecasts over time (see Sec. 3.3). This progressive generation is actually closer to reality because the multi-modality of motion should be more pronounced after further time. However, in this case, APD cannot reflect the diversity well, since many predictions will share the same previous segment. Therefore, we only examine the diversity of the *last pose* (FPD), as the last pose should not be the same.

Moreover, we introduce three *tailored* metrics to evaluate specific aspects of predicted motion, which correspond to the unique challenges in multi-person motion forecasting as discussed in the main paper. (a) **Foot Skating Ratio** (Zhang et al., 2021): the average ratio of frames with foot skating. (b) **Trajectory Collision Ratio**: the average ratio of predictions that is considered to have collision (Kothari et al., 2020) between any two trajectories in the scene. (c) **Average Human Displacement**: Average displacement of the predicted human body between the last frame and the first frame, reflecting the properties of the predicted motion distribution.

# G  ADDITIONAL EXPERIMENTAL RESULTS

**Additional Quantitative Results.** We compare our method with MRT (Wang et al., 2021b) in Table G. We show that the improvement is significant by performing each experiment five times with different random seeds and reporting error bars. When the number of intents is one, equivalent to a deterministic setting, our method marginally outperforms MRT for all metrics on CMU-Mocap and also generalizes better on MuPoTS-3D. This suggests the *generality* of our approach, which is also advantageous for deterministic processes.

In Table C, we use lADE and lFDE to compare single-person fidelity, and we observe that our model also significantly outperforms the baseline.

Table G: Quantitative results (w/ error bar) of DuMMF with a CGAN and the baseline MRT (Wang et al., 2021b). The baseline and our models are trained only on CMU-Mocap, and are tested on CMU-Mocap (top) and MuPoTS-3D (bottom). With the same backbone, our DuMMF framework significantly outperforms MRT on the deterministic prediction, and provides more accurate and diverse predictions with more intents and predictions.

| Method, # of Intents | @t = 1s | | | @t = 2s | | | @t = 3s | | |
|---|---|---|---|---|---|---|---|---|---|
| | FPD ↑ | ADE ↓ | FDE ↓ | FPD ↑ | ADE ↓ | FDE ↓ | FPD ↑ | ADE ↓ | FDE ↓ |
| MRT (Wang et al., 2021b), N/A | N/A | $0.682 \pm 0.005$ | $1.127 \pm 0.006$ | N/A | $1.082 \pm 0.006$ | $1.765 \pm 0.021$ | N/A | $1.435 \pm 0.014$ | $2.449 \pm 0.057$ |
| + DuMMF (Ours), 1 | N/A | $\mathbf{0.670 \pm 0.005}$ | $\mathbf{1.117 \pm 0.014}$ | N/A | $\mathbf{1.061 \pm 0.009}$ | $\mathbf{1.709 \pm 0.028}$ | N/A | $\mathbf{1.380 \pm 0.017}$ | $\mathbf{2.293 \pm 0.062}$ |
| + DuMMF (Ours), 2 | $0.435 \pm 0.045$ | $0.668 \pm 0.008$ | $1.098 \pm 0.018$ | $0.831 \pm 0.097$ | $1.044 \pm 0.019$ | $1.645 \pm 0.045$ | $1.717 \pm 0.139$ | $1.342 \pm 0.033$ | $2.117 \pm 0.085$ |
| + DuMMF (Ours), 3 | $0.569 \pm 0.050$ | $0.664 \pm 0.007$ | $1.077 \pm 0.009$ | $1.106 \pm 0.101$ | $1.019 \pm 0.009$ | $1.566 \pm 0.029$ | $1.717 \pm 0.170$ | $1.286 \pm 0.020$ | $1.999 \pm 0.073$ |
| + DuMMF (Ours), 5 | $0.633 \pm 0.042$ | $0.658 \pm 0.011$ | $1.060 \pm 0.020$ | $1.316 \pm 0.111$ | $0.992 \pm 0.020$ | $1.475 \pm 0.033$ | $2.112 \pm 0.274$ | $1.226 \pm 0.025$ | $1.809 \pm 0.049$ |
| MRT (Wang et al., 2021b), N/A | N/A | $0.427 \pm 0.008$ | $0.749 \pm 0.012$ | N/A | $0.750 \pm 0.011$ | $1.377 \pm 0.033$ | N/A | $1.082 \pm 0.020$ | $2.059 \pm 0.062$ |
| + DuMMF (Ours), 1 | N/A | $\mathbf{0.411 \pm 0.008}$ | $\mathbf{0.727 \pm 0.014}$ | N/A | $\mathbf{0.713 \pm 0.012}$ | $\mathbf{1.270 \pm 0.035}$ | N/A | $\mathbf{0.996 \pm 0.027}$ | $\mathbf{1.820 \pm 0.069}$ |
| + DuMMF (Ours), 2 | $0.334 \pm 0.110$ | $0.416 \pm 0.003$ | $0.729 \pm 0.011$ | $0.639 \pm 0.233$ | $0.715 \pm 0.015$ | $1.253 \pm 0.049$ | $1.000 \pm 0.377$ | $0.990 \pm 0.036$ | $1.783 \pm 0.104$ |
| + DuMMF (Ours), 3 | $0.472 \pm 0.055$ | $0.410 \pm 0.007$ | $0.712 \pm 0.013$ | $0.986 \pm 0.121$ | $0.694 \pm 0.019$ | $1.175 \pm 0.068$ | $1.623 \pm 0.185$ | $0.938 \pm 0.041$ | $1.586 \pm 0.143$ |
| + DuMMF (Ours), 5 | $0.513 \pm 0.061$ | $0.405 \pm 0.007$ | $0.703 \pm 0.014$ | $1.122 \pm 0.150$ | $0.678 \pm 0.018$ | $1.139 \pm 0.046$ | $1.887 \pm 0.277$ | $0.905 \pm 0.034$ | $1.509 \pm 0.083$ |

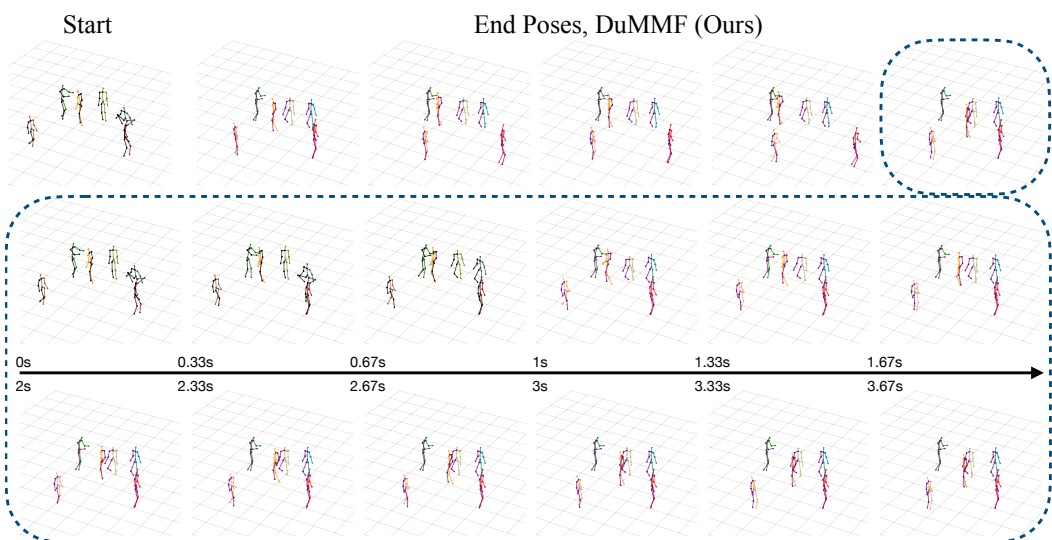

Figure A: Qualitative results on CMU-Mocap. We evaluate the generalizability and scalability of our model to predict 3-second motion on the constructed **6**-person motion test data. The top row is the historical pose, and the five end poses predicted by our model; and we show the sequence of one of the predictions in the bottom two rows (highlighted by the blue dashed box).

**Additional Comparisons on SoMoF Benchmark.** We investigate the performance of our framework on the Social Movement Prediction Challenge (SoMoF) (Adeli et al., 2020; 2021). We specifically use the 3DPW dataset (von Marcard et al., 2018), where we use labeled trajectories and poses (13-joint human skeleton), but we do not use videos of given scenes as input. We discard the multi-modal reconstruction loss since the 3DPW provided by SoMoF benchmark is relatively small. In Table E, we provide results on our implemented deterministic prediction and compare with baselines directly reported from the leaderboard on the SoMoF benchmark. In Table F, we use SoMoF benchmark for stochastic multi-person forecasting. Note that ADE and FDE require access to ground truth data, which is not publicly available from SoMoF benchmark. Thus, we report the results on validation set. We observe a similar performance as on CMU-Mocap (CMU) and MuPoTS-3D (Mehta et al., 2018), specifically, using DuMMF, the model has significantly better accuracy and diversity in stochastic multi-person forecasting.

**Additional Comparison with Single-Person Forecasting Methods.** In Table H, we provide a comparison by applying a stochastic single-person forecasting baseline. For a fair comparison, we formulate this single-person forecasting architecture (named SRT) with a transformer encoder-decoder adapted from MRT (Wang et al., 2021b), where we modified the encoder and decoder to be individually independent. Table H illustrates that (1) the modeling of social interactions is crucial, since the single-person forecasting baseline has much worse performance; (2) the improvement from

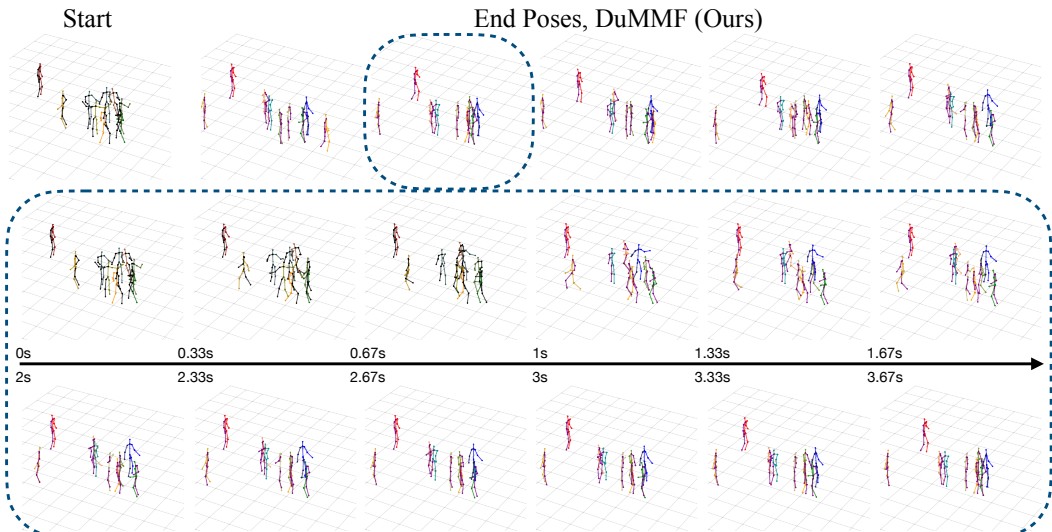

Figure B: Qualitative results on CMU-Mocap. We evaluate the generalizability and scalability of our model to predict 3-second motion on the constructed **9**-person motion test data. The top row is the historical pose, and the five end poses predicted by our model; and we show the sequence of one of the predictions in the bottom two rows (highlighted by the blue dashed box).

Table H: Quantitative comparison on CMU-Mocap between our DuMMF based on MRT (Wang et al., 2021b) and a single-person forecasting baseline adapted from MRT. The number of intents is set to 5 for stochastic forecasting on 3-person. For DuMMF with SRT, we use the variant of "w/o Separation" to skip the social interaction modeling in DuMMF, making it a full single-person prediction baseline.

| Architecture | Predictor | Diversifier | Variants | @t = 1s | | | @t = 2s | | | @t = 3s | | |
|---|---|---|---|---|---|---|---|---|---|---|---|---|
| | | | | FPD ↑ | ADE ↓ | FDE ↓ | FPD ↑ | ADE ↓ | FDE ↓ | FPD ↑ | ADE ↓ | FDE ↓ |
| Transformer | SRT | CGAN | N/A | N/A | 0.681 | 1.125 | N/A | 1.082 | 1.765 | N/A | 1.427 | 2.438 |
| | MRT (Wang et al., 2021b) | | | 0.282 | 0.662 | 1.086 | 0.662 | 1.023 | 1.567 | 1.199 | 1.287 | 1.968 |
| | SRT | CGAN+DuMMF (Ours) | w/o Separation | 0.291 | 0.677 | 1.110 | 0.676 | 1.056 | 1.624 | 1.166 | 1.328 | 2.021 |
| | MRT (Wang et al., 2021b) | | Full | **0.716** | **0.658** | **1.053** | **1.435** | **0.993** | **1.472** | **2.206** | **1.232** | **1.823** |

our DuMMF for the multi-person forecasting model is much higher than that for the single-person forecasting model.

**Additional Ablation on Impact of Learning Objectives.** In Table I, we evaluate the impact of each loss term within the dual-level framework. In general, using all loss functions yields the best results, since its results are either the best or the second best. We observe that local and global discriminators not only make predictions more accurate, but also more diverse. Note that the reconstruction losses $\mathcal{L}_{lR}$ and $\mathcal{L}_{gR}$ optimize only the most accurate prediction of all the outputs. It is important to provide supervision for other predictions that are not the best. We observe that limb loss $\mathcal{L}_L$ is crucial, as it is the only loss function that provides supervision for *all outputs*. The multi-modal reconstruction loss also has a large performance impact, since it provides supervision for more than one output.

**Additional Analysis of the Number of Discrete Latent Codes.** Note that the number of discrete latent codes is not restricted to the same number as the number of predictions per second. We chose

Table I: Ablation study of our DuMMF model on CMU-Mocap using skeletal representations. We report the accuracy and diversity with error bar for root and pose respectively. The results show the impact of different learning objectives. Best results are bolded, and next best results are underlined.

| $\mathcal{L}_{lR}$ | $\mathcal{L}_L$ | $\mathcal{L}_{lGAN}$ | $\mathcal{L}_{mmR}$ | $\mathcal{L}_D$ | $\mathcal{L}_{gGAN}$ | rootFPD ↑ | poseFPD ↑ | rootADE ↓ | rootFDE ↓ | poseADE ↓ | poseFDE ↓ |
|---|---|---|---|---|---|---|---|---|---|---|---|
| | ✓ | ✓ | ✓ | ✓ | ✓ | $\underline{0.251 \pm 0.025}$ | $\mathbf{0.541 \pm 0.047}$ | $0.748 \pm 0.010$ | $0.919 \pm 0.017$ | $0.247 \pm 0.007$ | $0.401 \pm 0.013$ |
| ✓ | | ✓ | ✓ | ✓ | ✓ | $0.242 \pm 0.069$ | $0.437 \pm 0.146$ | $0.784 \pm 0.029$ | $0.983 \pm 0.075$ | $0.274 \pm 0.015$ | $0.459 \pm 0.030$ |
| ✓ | ✓ | | ✓ | ✓ | ✓ | $0.241 \pm 0.026$ | $0.536 \pm 0.055$ | $\mathbf{0.735 \pm 0.005}$ | $0.897 \pm 0.005$ | $0.250 \pm 0.004$ | $0.407 \pm 0.014$ |
| ✓ | ✓ | ✓ | | ✓ | ✓ | $0.218 \pm 0.074$ | $0.468 \pm 0.186$ | $0.749 \pm 0.011$ | $0.920 \pm 0.017$ | $0.254 \pm 0.012$ | $0.418 \pm 0.038$ |
| ✓ | ✓ | ✓ | ✓ | | ✓ | $0.240 \pm 0.032$ | $0.537 \pm 0.067$ | $\underline{0.738 \pm 0.008}$ | $0.900 \pm 0.010$ | $\mathbf{0.241 \pm 0.010}$ | $\underline{0.393 \pm 0.021}$ |
| ✓ | ✓ | ✓ | ✓ | ✓ | | $0.232 \pm 0.058$ | $0.506 \pm 0.111$ | $0.740 \pm 0.004$ | $0.905 \pm 0.012$ | $0.247 \pm 0.008$ | $0.402 \pm 0.028$ |
| ✓ | ✓ | ✓ | ✓ | ✓ | ✓ | $\mathbf{0.254 \pm 0.037}$ | $\underline{0.538 \pm 0.071}$ | $0.738 \pm 0.007$ | $\mathbf{0.897 \pm 0.010}$ | $\underline{0.245 \pm 0.008}$ | $\mathbf{0.391 \pm 0.015}$ |

Table J: Ablation study of the number of discrete latent codes on CMU-Mocap using skeletal representations. For producing 5 predictions per second, we observe that both the accuracy and diversity of predictions decrease significantly as the number of discrete intents increases.

| # of Intents | @t = 1s | | | @t = 2s | | | @t = 3s | | |
|---|---|---|---|---|---|---|---|---|---|
| | FPD ↑ | ADE ↓ | FDE ↓ | FPD ↑ | ADE ↓ | FDE ↓ | FPD ↑ | ADE ↓ | FDE ↓ |
| 5 | **0.716** | **0.658** | **1.053** | **1.435** | **0.993** | **1.472** | **2.206** | **1.232** | **1.823** |
| 6 | 0.498 | 0.667 | 1.087 | 1.086 | 1.033 | 1.578 | 1.814 | 1.300 | 1.971 |
| 7 | 0.465 | 0.671 | 1.093 | 0.960 | 1.032 | 1.582 | 1.502 | 1.298 | 1.986 |
| 9 | 0.263 | 0.669 | 1.087 | 0.565 | 1.027 | 1.578 | 0.941 | 1.295 | 2.028 |
| 15 | 0.231 | 0.678 | 1.127 | 0.532 | 1.076 | 1.704 | 0.966 | 1.384 | 2.209 |

them to be the same for simplicity and a better trade-off between prediction performance and training efficiency. We find that this setup also achieved the best performance. To ensure that all predictions come from different discrete codes, the number of discrete latent codes should not be less than the number of predictions. In Table J, we provide an ablation study when the number of predictions per second is 5. If the number of discrete intents is greater than 5, we randomly select a discrete code without replacement (excluding any of the previously selected intents) to generate a prediction. We observe that both the accuracy and diversity of the predictions decrease as the number of intents increases. An explanation of such behavior could be: When the number of discrete latent codes is the same as the number of predictions, each code is *explicitly and fully optimized* for a particular prediction, leading to the best prediction accuracy and diversity; whereas, when the number of intents increases, the random selection strategy may hurt performance, as the probability of selecting the best five discrete codes decreases. We hypothesize that a better training and selection strategy might improve performance with more discrete codes.

# H  GENERALIZABILITY AND SCALABILITY: EVALUATION ON MORE-PERSON SCENARIOS

**Datasets with More People per Scene.** In the main paper, we constructed multi-person motion data with 3 people per scene. Here, we construct more challenging datasets with a significantly increased number of people; this also increases the complexity of social interactions. Specifically, we first sample 2-person and 1-person motion data from CMU-Mocap (CMU), and then compose them together. We use handcraft rules to filter out scenes with trajectory collisions. Instead of retraining the model in the novel setting with more people per scene, we *directly evaluate the model trained on 3-person motion data*, and test if our model is able to scale to scenarios with more people.

**Qualitative Results.** Similar to Figure 3 of the main paper, we provide visualizations for 6-person and 9-person scenarios in Figure A and Figure B of the end poses of predictions of 3 second. We randomly select a prediction and visualize the corresponding full motion sequence as well. Note that our model is trained on only 3-person data without any fine-tuning on more-person data, but it still performs well on more-person data, suggesting the *generalizability* and *scalability* of our approach.

