# OpenReview forum: "Stochastic Multi-Person 3D Motion Forecasting"
_ICLR.cc/2023/Conference — ICLR 2023 notable top 25%_

### Official Review · Reviewer_UYRH · 2022-10-22

**Confidence:** 4
**Correctness:** 4
**Technical Novelty And Significance:** 2
**Empirical Novelty And Significance:** 4
**Recommendation:** 8

**Clarity, Quality, Novelty And Reproducibility:**

The clarity and quality of the paper is good. And based on the information provided in the paper, I guess the framework is reproducible. The novelty is not substantial but acceptable considering good results.

**Strength And Weaknesses:**

Strengths:

1- The idea and motivation are clear and the submission is well presented, 2- a comprehensive literature review with relevant literature and research topics is covered 3- extensive experiments on different dataset and one related benchmark is provided, 4- the method achieved SOTA results on these datasets

Weakness:

1-considering most generative models are well studied in multi agent trajectory forecasting, the paper’s technical contribution is yet relatively limited compared to the literature (though the problem here is not identical and the overall framework is yet different)

 2- a better qualitative visualisation of results would be helpful for better understanding of the framework behaviour

**Summary Of The Paper:**

This paper focuses on the problem of multi-person 3D motion forecasting. The proposed framework uses a dual-level transformer-based generative model, encoding the local (per person) and global (multi-person) motion histories to decode each individual future body motion. The framework has been comprehensively evaluated on different datasets such as CMU-Mocap, 3DPW and MuPoTS-3D and also a benchmark, SoMoF, and demonstrates its superior performance against SOTA frameworks.


**Summary Of The Review:**

Although the technical contribution is not substantial, I think the paper has a good quality in the terms of presentation, experiments and results. Considering these aspects, I support the paper to be accepted in this venue. However I have some minor comments

1-  on the presentation of qualitative results: it can barely understand anything from Figs 3-5. The Errors and the good results could be highlighted. The figures are also too small to distinguish anything meaningful.

2- I am not sure if I found out how did the predicted samples from generative models are evaluated during inference? I think eventually one sample from a distribution should represent the final output and this should be decided independently from evaluating them against ground truth. Do you calculate the mode of the generated samples? If so how?

3-  while the model is compared with most recent conference papers published on SoMoF benchmark, it would be great if the authors can comment on other published frameworks (eg workshop ones) which use very simple and non-transformer based frameworks with superior results. For comparison purpose, It would best if the authors make their results on SoMoF leaderboard published .

---

> ### Author Response · Authors · 2022-11-17
> **Response to Reviewer UYRH (part 2)**
>
> *W4: Comparison and discussion on SoMoF benchmark*
>
> We agree with the reviewer that many methods on the SoMoF leaderboard achieve strong performance with simple and non-transformer-based frameworks, such as [Ref4, Ref5, Ref6]. Our hypothesis is that this is because the dataset provided by the SoMoF benchmark is relatively small, as mentioned in the paragraph "Additional Comparisons on SoMoF Benchmark" in Sec. G of the Appendix. Specifically, futuremotion_ICCV21 [Ref4] develops a simple but effective framework based on a combination of graph convolutional networks and multiple motion optimization techniques. DViTA [Ref5] decouples the human pose into a trajectory and local pose and employs a VAE to learn a representation of the local pose dynamics. LSTMV_LAST [Ref6] proposes a sequence-to-sequence LSTM via keypoint-based representation. We will include all the discussion in the revision.
>
> The evaluation of our DuMMF framework on SoMoF is based on MRT [Ref3] which is a transformer-based predictor backbone and achieves the best results on our major datasets. However, our DuMMF framework is generic and agnostic to the predictor architectures. Throughout our paper, we evaluated both transformer-based architecture and non-transformer-based architecture as backbones, as explained in the "Baselines" paragraph of Sec. 4 in the main paper,
>
> Also, as suggested by the reviewer, we will make our deterministic prediction results on the SoMoF leaderboard published upon acceptance following the anonymity policy.
>
> **References:**
>
> [Ref1] Ye Yuan and Kris Kitani. DLow: Diversifying latent flows for diverse human motion prediction. ECCV, 2020.
>
> [Ref2] Wei Mao, Miaomiao Liu, and Mathieu Salzmann. Generating smooth pose sequences for diverse human motion prediction. CVPR, 2021.
>
> [Ref3] Jiashun Wang, Huazhe Xu, Medhini Narasimhan, and Xiaolong Wang. Multi-Person 3D Motion Prediction with Multi-Range Transformers. NeurIPS, 2021.
>
> [Ref4] Chenxi Wang, Yunfeng Wang, Zixuan Huang, and Zhiwen Chen. Simple Baseline for Single Human Motion Forecasting. ICCVW, 2021.
>
> [Ref5] Saeed Saadatnejad, Behnam Parsaeifard, Yuejiang Liu, Taylor Mordan, and Alexandre Alahi. Learning Decoupled Representations for Human Pose Forecasting. ICCVW, 2021.
>
> [Ref6] Armin Saadat, Nima Fathi, Saeed Saadatnejad, and Alexandre Alahi. Towards Human Pose Prediction using the Encoder-Decoder LSTM. ICCVW, 2021.
>
> [Ref7] Tim Salzmann, Boris Ivanovic, Punarjay Chakravarty, and Marco Pavone. Trajectron++: Dynamically-feasible trajectory forecasting with heterogeneous data. ECCV, 2020.
>
> [Ref8] Chenxin Xu, Maosen Li, Zhenyang Ni, Ya Zhang, and Siheng Chen. GroupNet: Multiscale Hypergraph Neural Networks for Trajectory Prediction With Relational Reasoning. CVPR, 2022.

---

> ### Author Response · Authors · 2022-11-17
> **Response to Reviewer UYRH (part 1)**
>
> We thank the reviewer for the positive comments. We are glad that the reviewer finds our motivation clear, the literature review comprehensive, the experiments extensive, and the results strong. Below we address all the concerns.
>
> *W1: Technical contribution with respect to generative models in multi-agent trajectory forecasting*
>
> We agree with the reviewer that generative models are well studied in multi-agent trajectory forecasting. However, compared with multi-agent trajectory forecasting, our task of multi-person 3D motion forecasting poses **substantially increased complexity with multi-faced new challenges**. Specifically, we need to **simultaneously** address 1) single-person pose fidelity, 2) consistency of pose and trajectory, 3) social interactions between poses, and 4) overall diversity of motions.
>
> Therefore, these new challenges call for a novel solution to stochastic multi-person 3D motion forecasting. The existing work on trajectory forecasting does not focus on alleviating the burden on generative models from learning all the aforementioned objectives simultaneously. As shown in Table 2 of the original submission and Table D of the Appendix, generative models struggle to handle all learning objectives at both global level and local level simultaneously and fail to produce diverse and accurate predictions; on the contrary, by leveraging dual-level modeling and discrete social intents, our DuMMF has achieved very significant improvements in both diversity and accuracy. Notably, our DuMMF is a general framework -- it can be combined with different predictor architectures and shows consistent improvements. For example, with our DuMMF and the MRT predictor architecture [Ref3], the two-person forecasting result is improved by up to 14% on accuracy and up to 69% on diversity.
>
>
> *W2: Qualitative visualization needs more explanation*
>
> We thank the reviewer for this great suggestion. We would like to kindly refer the reviewer to the demo video which we provided in the original supplementary material. In this video, we showed larger visualizations, highlighted the important regions, and complemented them with additional explanations.
>
> In addition, as suggested by the reviewer, in the revision we will highlight the important regions in Figures 3-5 and enlarge the figures for better understanding.
>
> *W3: Evaluation method during inference*
>
> In the "Metrics" paragraph of Sec. 4 in the original submission, we explained the major metrics used to evaluate our method during inference. Moreover, we provided additional detail in Sec. F of the Appendix and included a summary of the metrics in Table B of the Appendix.
>
> These complementary metrics evaluate prediction accuracy and diversity. For accuracy measurement (including ADE and FDE), we follow the Best-of-N (BoN) evaluation, which has been widely adopted in the literature on stochastic single-person 3D pose forecasting [Ref1, Ref2] and stochastic multi-person trajectory forecasting [Ref7, Ref8]. For example, for ADE, we generate a set of predictions and calculate the distance between the ground truth and the prediction *closest* to the ground truth. Such Best-of-N (BoN) evaluation is different from evaluating the most likely sample.

---

### Official Review · Reviewer_r2ic · 2022-10-25

**Confidence:** 3
**Correctness:** 3
**Technical Novelty And Significance:** 3
**Empirical Novelty And Significance:** 2
**Recommendation:** 8

**Clarity, Quality, Novelty And Reproducibility:**


Clarity: the paper is clear and reads well
Quality: results are of high quality, but could have been more convincing with multiperson datasets
Originality: other work deals with social modules, this is the first dealing with social aspects



**Strength And Weaknesses:**

strenghts:
- The model is quite simple, it is based on a generative model which based on two steps: the first one aims to predict the single-person pose in the future, the second one aims to analize the social interactions among people in order to refine the single-person pose.
- the model is able to predict interactions between people and can learn for example how to react to collisions.
- the paper opens up to new tasks and problems, it can introduce a new branch of pose forecasting leading to new models capable of better understandings of human behaviours.
- details on implementation of the code are well described in supplementary material

weaknesses:
- Metrics used in my opinion aren't easy to understand from a practical point of view. It is difficult to have an idea on how the model can perform in terms of mm of error.
I suggest to use metrics like MPJPE or MAE to better understand the quantitative performances of the model.
- improving models with social aspect of the human movement can be really effective for the model performances. I would like to see an ablative study on how this model can improve performances compared to standard  single person forecasting models.
My doubt about this system is that performances compared to sota single person forecasting models would be lower, on standard datasets that offer even multi person poses (CMU panoptic, 3DPW).

**Summary Of The Paper:**

The paper presents a novel architecture to performe multi-person 3D pose forecasting. The importance of pose forecasting is well known, useful in quite complex situations like autonomous driving.
The model is quite simple but effective, taking into account an important variable of the human behaviour which is the context and the others in the environment.
The paper is integrated with an extensive experimental section.



**Summary Of The Review:**

The model is quite simple but effective, taking into account an important variable of the human behaviour which is the context and the others in the environment. It could have been ok to show some experiments on single pose forecasting too

---

> ### Author Response · Authors · 2022-11-17
> **Response to Reviewer r2ic (part 2)**
>
> *W2.2: Other datasets: CMU panoptic, 3DPW*
>
> We followed MRT [Ref3] to perform the evaluation on motion capture datasets: CMU-Mocap and MuPoTS-3D. Note that, to evaluate in more-person scenarios, we also manually synthesized scenes based on CMU-Mocap with, for example, 9 people, as shown in Figure 5 of the original submission.
>
> In addition, we further evaluated the performance of our framework on the Social Movement Prediction Challenge (SoMoF) which is based on the 3DPW dataset, in Sec. G of the original Appendix. And indeed, from the SoMoF leaderboard, we observe that some simple single-person forecasting methods achieve strong performance on multi-person forecasting, which we believe is due to the fact that SoMoF provides a very small dataset, and such a small dataset may not be able to support the learning of social interactions. However, we would like to point out that even in this small dataset, our DuMMF framework still facilitates multi-person prediction architectures, as shown in Table F of the Appendix.
>
> We thank the reviewer for suggesting CMU Panoptic, which contains a very large number of people in some sequences. We believe that our current evaluation validates the effectiveness of our approach, though including the evaluation on CMU Panoptic would be of interest. We are currently experimenting on CMU Panoptic, which takes time since a different skeleton representation than the openpose format is used, and we need to perform heavy preprocessing including generating multimodal ground truth, and retrain the model. We will include the result in the final version.
>
> **References:**
>
> [Ref1] Ye Yuan and Kris Kitani. DLow: Diversifying latent flows for diverse human motion prediction. ECCV, 2020
>
> [Ref2] Wei Mao, Miaomiao Liu, and Mathieu Salzmann. Generating smooth pose sequences for diverse human motion prediction. CVPR, 2021
>
> [Ref3] Jiashun Wang, Huazhe Xu, Medhini Narasimhan, and Xiaolong Wang. Multi-Person 3D Motion Prediction with Multi-Range Transformers. NeurIPS, 2021
>
> [Ref4] Wei Mao, Miaomiao Liu, Mathieu Salzmann, and Hongdong Li. Learning Trajectory Dependencies for Human Motion Prediction. ICCV, 2019.
>
> [Ref5] Vida Adeli, Ehsan Adeli, Ian Reid, Juan Carlos Niebles, and Hamid Rezatofighi. Socially and contextually aware human motion and pose forecasting. RA-L, 2020.

---

> ### Author Response · Authors · 2022-11-17
> **Response to Reviewer r2ic (part 1)**
>
> We thank the reviewer for the positive comments. We are glad that the reviewer finds our method simple but effective and strong, the experiments extensive, and that our paper opens up new tasks and problems. Below we address all the concerns.
>
> *W1: Use of Metrics*
>
> We thank the reviewer for suggesting using MPJPE or MAE as the metric. For accuracy measurements, we mostly adopted from the literature on stochastic single-person motion forecasting [Ref1, Ref2]. In fact, ADE (which we used) is the Best-of-N (BoN) version of MPJPE or MAE for stochastic prediction (Please refer to the formula in Table B of the original Appendix). If the number of predictions is 1, *i.e.*, deterministic prediction, ADE is equivalent to MPJPE -- we also provided the results of deterministic prediction with such a metric in all tables in the original submission. We will clarify this in the revision.
>
> Please refer to the detailed explanation on the used metrics in the original Appendix -- in addition to the description of metrics in Sec. 4, we systematically presented complementary evaluation metrics and provided more details in Section F and Table B of the original Appendix.
>
> *W2.1: Comparison to the sota single-person forecasting model*
>
> First, we would like to clarify that our goal is to address the challenge of extending deterministic multi-person forecasting to stochastic multi-person forecasting, where we show that our approach is general and provides significant enhancements to all current deterministic multi-person forecasting.  Comparison with the single-person motion forecasting method on multi-person datasets is not our focus and has already been studied in [Ref3, Ref5]. For example, in MRT [Ref3], considering only single-person motion using [Ref4] without modeling social interaction led to worse performance.
>
> Here, per the reviewers' request, we provide a comparison by applying a stochastic single-person forecasting baseline. For a fair comparison, we formulate this single-person forecasting architecture (named as SRT) with a transformer encoder-decoder adapted from MRT [Ref3], where we modified both encoder and decoder to be individual independent. The table below illustrates that (1) the modeling of social interaction is crucial, since the single-person forecasting baseline has much worse performance; (2) the improvement from our DuMMF for the multi-person forecasting model is much higher than that for the single-person forecasting model.
>
>
> |Method|FPD@1s $\uparrow$|ADE@1s $\downarrow$|FDE@1s $\downarrow$|FPD@2s $\uparrow$|ADE@2s $\downarrow$|FDE@2s $\downarrow$|FPD@3s $\uparrow$|ADE@3s $\downarrow$|FDE@3s $\downarrow$|
> |-|-|-|-|-|-|-|-|-|-|
> |SRT+CGAN|0.260|0.684|1.128|0.633|1.092|1.767|1.149|1.411|2.264|
> |MRT+CGAN|0.282|0.662|1.086|0.662|1.023|1.567|1.199|1.287|1.968|
> |SRT+**DuMMF** (Ours)|0.351|0.674|1.115|0.843|1.075|1.730|1.474|1.395|2.250|
> |MRT+**DuMMF** (Ours)|**0.716**|**0.658**|**1.053**|**1.435**|**0.993**|**1.472**|**2.206**|**1.232**|**1.823**|
>
> |Method@3s|rootFPD $\uparrow$|rootADE $\downarrow$|rootFDE $\downarrow$|poseFPD $\uparrow$|poseADE $\downarrow$|poseFDE $\downarrow$|
> |-|-|-|-|-|-|-|
> |SRT+CGAN|0.126|0.773|0.957|0.291|0.299|0.524|
> |MRT+CGAN|0.167|0.748|0.921|0.303|0.260|0.433|
> |SRT+**DuMMF** (Ours)|0.142|0.756|0.945|0.374|0.293|0.515|
> |MRT+**DuMMF** (Ours)|**0.249**|**0.734**|**0.898**|**0.562**|**0.243**|**0.390**|

---

### Official Review · Reviewer_Lood · 2022-10-25

**Confidence:** 5
**Correctness:** 4
**Technical Novelty And Significance:** 3
**Empirical Novelty And Significance:** 3
**Recommendation:** 8

**Clarity, Quality, Novelty And Reproducibility:**

Details are in [Weaknesses].

Clarity:
The clarity is good.

Quality:
Clear-written and well-organized.

Novelty:
Motivation is strong and novelty is good as well.

Reproducibility:
The training procedure is clear, and authors also provide the codes.

**Strength And Weaknesses:**

**Strengths**:
Overall, it is a good paper with strong motivation. The method is clearly introduced and the experiments are sufficient and convincing.


**Weaknesses**:
1. In the proposed method, why are the latent codes of the social interactions defined as the same? What's the insight here? In my opinion, the interactions are also dynamically changing during the observations, which is also pointed in [1,2]. Some explanations should be included here. In addition, why consider social interactions on a global level? How about the local social interactions?

2. In Sec. 3.1, Local-Level Modeling, $N$ different intents $z_{i}$ are extracted to generate independent future movements. Does $N$ represent the total number of people in one sequence or one batch or the whole dataset? I presume it is the number of people in one sequence or batch, then $N$ must vary in different sequences or batches. How to deal with this situation? More details should be included here.


3. Missing related works, some latest multi-person forecasting trajectory/prediction [1, 2, 3]. Since this work focuses on multi-person motion prediction, I think part of the discussion in Sec. B of the Appendix should be summarized in Related Work Section to highlight the differences between the proposed method and existing ones.

[1] Evolvegraph: Multi-Agent Trajectory Prediction with Dynamic Relational Reasoning. NeurIPS2020
[2] Tra2tra: Trajectory-to-Trajectory Prediction with A Global Social Spatial-Temporal Attentive Neural Network. ICRA 2021
[3] GroupNet: Multiscale Hypergraph Neural Networks for Trajectory Prediction With Relational Reasoning. CVPR2022
[4] Adaptive Trajectory Prediction via Transferable GNN. CVPR2022
[5] Stochastic Trajectory Prediction via Motion Indeterminacy Diffusion. CVPR2022


**Summary Of The Paper:**

In this paper, authors propose a dual-level generative model, considering the individual movements and social interactions. The proposed method achieves good performance on CMU-Mocap, MuPoTS-3D, and SoMoF benchmarks.

**Summary Of The Review:**

I list my concerns in [Weaknesses], I am happy to discuss and increase the rating if my concerns are addressed.

---

> ### Author Response · Authors · 2022-11-17
> **Response to Reviewer Lood (part 2)**
>
> *W3:* *Discussion on latest multi-person trajectory forecasting*
>
> We thank the reviewer for the suggested reference. We will cite all the papers and include the discussion below in the revision. Also, in our original submission, due to limited space, some additional discussion on related work was included in Sec. B of the Appendix. As suggested by the reviewer, we will also include a summary of Sec. B in the main paper.
>
>
> Here we first summarize the papers suggested by the reviewer and then discuss their difference from our work. EvolveGraph [Ref1] introduces an effective dynamic mechanism to evolve interaction graphs, which can flexibly model dynamically changing interactions. Tra2tra [Ref2] introduces a global social spatio-temporal attention neural network to encode both spatial interactions and temporal features. GroupNet [Ref3] employs a multi-scale hypergraph neural network that models group-based interactions and facilitates more comprehensive relational reasoning. T-GNN [Ref4] introduces a transferable graph neural network that allows not only trajectory prediction but also domain alignment of potential distribution differences. MID [Ref5] employs a diffusion model to model the variation of indeterminacy for trajectory prediction.
>
> The difference is that these papers focus on multi-person trajectory forecasting, whereas our task of multi-person 3D motion forecasting **poses substantially increased complexity with multi-faced new challenges**. Specifically, we need to **simultaneously** address 1) single-person pose fidelity, 2) consistency of pose and trajectory, 3) social interactions between poses, and 4) overall diversity of motions. And these new challenges call for a novel solution to stochastic multi-person 3D motion forecasting. The papers we discussed above do not focus on alleviating the burden on generative models from learning all the aforementioned objectives simultaneously. As shown in Table 2 of the original submission and Table D of the Appendix, generative models struggle to handle all learning objectives at both global level and local level simultaneously and fail to produce diverse and accurate predictions; by contrast, our DuMMF has achieved very significant improvements on both diversity and accuracy.
>
> **References:**
>
> [Ref1] Jiachen Li, Fan Yang, Masayoshi Tomizuka, and Chiho Choi. EvolveGraph: Multi-Agent Trajectory Prediction with Dynamic Relational Reasoning. NeurIPS, 2020.
>
> [Ref2] Yi Xu, Dongchun Ren, Mingxia Li, Yuehai Chen, Mingyu Fan, and Huaxia Xia. Tra2tra: Trajectory-to-Trajectory Prediction with A Global Social Spatial-Temporal Attentive Neural Network. ICRA, 2021.
>
> [Ref3] Chenxin Xu, Maosen Li, Zhenyang Ni, Ya Zhang, and Siheng Chen. GroupNet: Multiscale Hypergraph Neural Networks for Trajectory Prediction With Relational Reasoning. CVPR, 2022.
>
> [Ref4] Yi Xu, Lichen Wang, Yizhou Wang, and Yun Fu. Adaptive Trajectory Prediction via Transferable GNN. CVPR, 2022.
>
> [Ref5] Tianpei Gu, Guangyi Chen, Junlong Li, Chunze Lin, Yongming Rao, Jie Zhou, and Jiwen Lu. Stochastic Trajectory Prediction via Motion Indeterminacy Diffusion. CVPR, 2022.

---

> > ### Comment · Reviewer_Lood · 2022-11-29
> > **Post-rebuttal**
> >
> > Thank you for answering my question and sorry for the late response. After reading your answers and other reviews, as well as your revised submission, I will increase my score to 8 as most of my concerns are addressed.

---

> ### Author Response · Authors · 2022-11-17
> **Response to Reviewer Lood (part 1)**
>
> We thank the reviewer for the positive comments. We are glad that the reviewer finds our motivation strong, the method clear, and the experiments sufficient and convincing. Below we address all the concerns.
>
> *W1.1: Social intents are defined as the same latent codes across individuals*
>
> We thank the reviewer for the suggestion of including an explanation here. Intuitively, a necessary condition for modeling interactions is the sharing of information between different individuals. Defining the same code as a social intent to be shared among different individuals is a simple yet effective way to share information. An alternative, more sophisticated strategy could be an attention mechanism or similarity measure. However, we found that our simple sharing mechanism works well in practice.
>
> In addition, keeping social intents the same does not discard the flexibility of our predictive framework; for example, the network may learn to use different components of the same intent code based on different observations of different individuals. Still, we agree with the reviewer that more explicitly considering the interaction pattern to be dynamically evolving [Ref1, Ref2] throughout the future time steps might lead to better performance, which we leave as interesting future work.
>
> *W1.2: Considering social interactions on a global level*
>
> Similarly, the network's ability to use different components of the same code, depending on different observations of different individuals, suggests that this global approach does not completely exclude the consideration of localities. However, a completely local representation for social intents might not make sense, since our definition of social intents requires the involvement of motion from different individuals, *i.e.*, the sharing of information between different individuals.
>
> *W2:* *Number of intents and number of people*
>
> In "Basic Generative Modeling Framework" of Sec. 3.1, we provided the definition of $\\{\mathbf X_n\\}\_{n=1}^N, \\{\mathbf Y_n\\}\_{n=1}^N$, where $N$ represents the total number of people in one sequence. As illustrated in Figure 2\(c\) and Sec. 3.2, intents are sampled. We do not have a limit on the number of times we can sample from a Gaussian distribution (for continuous intents) or from a discrete distribution (for discrete intents). That means, we just need to sample more times if we have more individuals -- doing so deals with the situation when $N$ varies in different sequences.

---

### Official Review · Reviewer_Eu4r · 2022-10-26

**Confidence:** 4
**Correctness:** 2
**Technical Novelty And Significance:** 2
**Empirical Novelty And Significance:** 2
**Recommendation:** 8

**Clarity, Quality, Novelty And Reproducibility:**

The work is clear except the latent coding part.
Novelty is limited. See above discussion.

**Strength And Weaknesses:**

Introduction level discussion:

“stochastic forecasting of human trajectories in crowds …”
Advanced a lot with modeling the social interactions and multiple works showed this.
For example, Trajectron++… etc. Authors should re-visit this statement in the introduction.

The “multi-person 3D motion forecasting” task already exists. See “Multi-Person 3D Motion Prediction with Multi-Range Transformers” . Claiming it is “novel” because the word “stochastic” is not accurate. Motion forecasting/ prediction is a generic task that can be solved deterministically or stochastically.

The concept of local and global dealing with motion forecasting already existed in prior works such as Trajectron++ and follow works… etc.


Method discussion:
The discrete intent is positive of this work. Though, is not sampling the same intent for all individuals without conditioning on the individual trajectory might lead to mode collapse?

Results discussion:
- Are the ADE/FDE metrics best of N (B-o-N)?
- How many samples used in calculating the performance? Does the results vary with respect to the number of samples?
- When evaluating 3D person motion prediction we are expected  to use the 3D path error and 3D pose error to understand the accuracy of the joints. Please see “Long-term human motion prediction with scene context “ and later works for these metrics.
- What is the divergence of the prediction between 1,2,3 seconds?
- Ablation study is positive.


**Summary Of The Paper:**

The work presents a deep GAN model to predict the 3d motion of humans in a social interaction. It introduces a dynamic intent model that models the interaction between the representation of the individual motions.

**Summary Of The Review:**

The work is good. It need to re-visit several claims in the introduction part.
The method needs a better illustration.
The evaluation misses a core metrics and clarification of how the comparison versus deterministic models was constructed.

---

> ### Author Response · Authors · 2022-11-17
> **Response to Reviewer Eu4r (part 2)**
>
> *Q4: Local and global dealing with forecasting in Trajectron++*
>
> We thank the reviewer for the suggested reference, and we will cite and discuss it in the revision as below.
>
> Our work is different from Trajectron++ [Ref1] in three important ways for local and global dealing with forecasting. (1) As mentioned before, our work is on 3D motion forecasting, while Trajectron++ is on trajectory forecasting. These are **different tasks**, and their local and global modeling means different aspects and handles different challenges. (2) The **concepts** of local and global dealing with forecasting are different. In Trajectron++, there is **no explicitly** global and local separated modeling, while surroundings that are close to the agent are specifically referred to as the local part. However, in our DuMMF, we have explicitly proposed a dual-level modeling, where the local level refers to the independent components of individual movements, and the global level refers to their interrelation, *i.e.*, the impact of social interaction on each individual movement. (3) The **specific modeling strategies** are different. Trajectron++ constructs a spatio-temporal scene graph where nodes represent individuals and edges represent their interactions, and global and local modeling is embedded in this graph structure. By contrast, the dual-level modeling in our DuMMF is not based on the architectural design, but on the generative modeling process, through simply switching the modes of latent codes of the generative model to represent different levels. Notably, our dual-level modeling is predictor-architecture agnostic; it can be combined with different predictor architectures and shows consistent improvements.
>
> **Results discussion:**
>
> *Q5: ADE and FDE are best of N (B-o-N)*
>
> Yes. As explained in the "Metrics" paragraph of Sec. 4 in the original submission (as well as Table B of the Appendix), ADE and FDE are the distance between the ground truth and the prediction *closest* to the ground truth, which are exactly the Best-of-N (BoN) measurements.
>
> *Q6: Performance w.r.t. the number of samples*
>
> In the original submission, we provided the performance w.r.t. the number of samples in Table 1. We have shown that our DuMMF framework provides more accurate and diverse predictions with more intents and predictions. Specifically, the number of predictions is determined by the number of social intents, and in Table 1 in 1 second we evaluated the number of samples from 1 to 5. Please refer to the "Inference" paragraph of Sec. 3.3 for more details.
>
> *Q7: 3D path error and 3D pose error*
>
> In the original submission, we **provided ADE, FDE, and FPD with alignment to evaluate the pose and trajectory separately**. Specifically, as shown in Table 3 and as explained in "Metrics" of Sec. 4, we explicitly mentioned that we disentangle the local pose and the global trajectory (path) of the motion, and measure their accuracy and diversity separately by defining the following metrics: rootADE, rootFDE, poseADE, poseFDE, rootFPD, and poseFPD.
>
> *Q8: Divergence of the prediction between 1,2,3 seconds*
>
> We are not quite sure what the divergence means here. If you are referring to the definition of prediction at 1, 2, or 3 seconds, in the original submission, we provided the explanation of the *progressive diversity* in the “Inference” paragraph of Sec. 3.3 and Figure 1\(c\), where we set the number of predictions to grow with the length of the prediction at 1, 2, and 3 seconds in the evaluation. If the reviewer could kindly elaborate more on this question, we are happy to further clarify.
>
> **References:**
>
> [Ref1] Salzmann et al. Trajectron++: Dynamically-feasible trajectory forecasting with heterogeneous data. ECCV, 2020.
>
> [Ref2] Mao et al. Learning Trajectory Dependencies for Human Motion Prediction. ICCV, 2019.
>
> [Ref3] Yuan et al. DLow: Diversifying latent flows for diverse human motion prediction. ECCV, 2020.
>
> [Ref4] Mao et al. Generating smooth pose sequences for diverse human motion prediction. CVPR, 2021.
>
> [Ref5] Dang et al. MSR-GCN: Multi-Scale Residual Graph Convolution Networks for Human Motion Prediction. ICCV, 2021.
>
> [Ref6] Adeli et al. Socially and contextually aware human motion and pose forecasting. RA-L, 2020.
>
> [Ref7] Wang et al. Multi-Person 3D Motion Prediction with Multi-Range Transformers. NeurIPS, 2021.
>
> [Ref8] Guo et al. Multi-person extreme motion prediction. CVPR, 2022.

---

> > ### Comment · Reviewer_Eu4r · 2022-11-29
> > **Thanks**
> >
> > Thanks for your response. I've decided to raise my score based on the rebuttal and other reviews.

---

> ### Author Response · Authors · 2022-11-17
> **Response to Reviewer Eu4r (part 1)**
>
> We thank the reviewer for the insightful comments. We are glad that the reviewer finds our work good, and the ablation study positive. Below we address all the concerns.
>
> **Introduction discussion:**
>
> *Q1: Re-visit the statement “stochastic forecasting of human trajectories in crowds …"*
>
> We thank the reviewer for the suggestion. Our original statement was "... has shown encouraging progress, but it only considers the motion and interaction at the trajectory level ..." We agree with the reviewer that modeling social interaction in trajectory forecasting has advanced a lot, e.g., Trajectron++ (Salzmann et al. 2020). On the other hand, we would like to clarify that this statement was to describe that modeling social interaction in our task of 3D motion forecasting is more challenging than that in trajectory forecasting.
>
> We now revise this sentence to "In addition, stochastic forecasting of human trajectories in crowds (Alahi et al., 2014; Kothari et al., 2020) has shown impressive progress on modeling social interaction, *e.g.*, with the use of attention models (Vemula et al., 2018; Sadeghian et al., 2019; Zhang et al. 2019) and spatial-temporal graph models (Huang et al. 2019; Mohamed et al. 2020; Yu et al. 2020). However, this task only considers the motion and interaction at the trajectory level. Modeling articulated motion involves richer human-like social interactions than trajectory forecasting which mainly focuses on trajectory collisions".
>
> *Q2: Novelty: difference from existing deterministic multi-person motion prediction*
>
> We respectfully disagree with the reviewer that extending deterministic motion prediction to stochastic motion prediction is not novel because motion prediction is a generic task and can be solved either deterministically or stochastically.
>
> Admittedly, **in trajectory forecasting**, deterministic and stochastic trajectory forecasting are sometimes discussed together. However, 3D human motion prediction is not a generic task that can be solved either deterministically or stochastically. In fact, **in most work on (single-person) human motion prediction, stochastic prediction [Ref3, Ref4] and deterministic prediction [Ref2, Ref5] have developed different literature with different methodologies** and different evaluation metrics. This is because they have to address significantly different challenges: stochastic motion forecasting addresses the mode collapse and the trade-off between diversity and human pose quality, which are new challenges that do not exist in deterministic motion forecasting.
>
> Regarding our extension from deterministic multi-person forecasting to stochastic multi-person forecasting, **at the problem level**, as mentioned by **reviewer r2ic**, our paper "opens up to new tasks and problems, it can introduce a new branch of pose forecasting leading to new models capable of better understanding of human behaviors." Specifically, our new task reveals real-world complexity with substantially *increased, multi-faced* challenges that need to *simultaneously* take into account single-person pose fidelity, consistency of pose and trajectory, social interactions between poses, and overall diversity of motions. In particular, in our stochastic scenario, such fidelity and interactions need to be both satisfied and diversified for all the predictions. Please refer to the supplementary video (00:00~00:43) for clearer visualizations of such complexities.
>
> Importantly, **at the methodology level**, our new challenging problem calls for novel solutions. As shown in Table 2 of the original submission and Table D of the Appendix, extending the state-of-the-art deterministic forecasting method [Ref7] with conventional generative modeling for stochastic forecasting fails to produce diverse and accurate predictions, substantially underperforming our method DuMMF.
>
> *Q3: Mode collapse from sampling the same intent for all individuals*
>
> First, we would like to clarify that the forecasting is **conditioned on both intents and individual historical motion**, as illustrated in Eq. 2~5 of the original submission. At the implementation level, our DuMMF framework is applied on top of several multi-person motion forecasting architectures [Ref6, Ref7, Ref8], as discussed in the "Baseline" paragraph of Sec. 4. Please refer to their papers for more details on how to predict multi-person motion conditioned on each person's historical motion. For example, MRT [Ref7] directly decodes the combination of the global state and the embedding from the individual's past motion to generate a corresponding future motion.
>
> From qualitative and quantitative results, *e.g.*, experiments in Table 2 and Table D of the Appendix, as well as in the supplementary video, we observe that using the same social intent does not result in a degradation in diversity (See the measurement of FPD), and therefore we believe that sampling the same intent does not lead to mode collapse.

---

### Author Response · Authors · 2022-11-19
**General Response to All Reviewers**

We thank all reviewers for their constructive and valuable comments and their interest in our approach.

Paper revision: We have uploaded a new version of the submission, in which we have included all the contents requested/suggested by the reviewers and edited the text accordingly. The main revision includes the following:

* In the second paragraph of Sec. 1, we have revised our statement on stochastic multi-person trajectory forecasting as suggested by **Reviewer Eu4r**.

* We have cited and supplemented the discussion of references. The additional discussion of trajectory-level forecasting work (suggested by **Reviewer Eu4r** and **Reviewer Lood**) and deterministic forecasting work on the SoMoF leaderboard (suggested by **Reviewer UYRH**) has been added in Sec. B of the Appendix.

* We have adjusted the discussion of related work in Sec. 2 of the main paper and Sec. B of the Appendix. We have highlighted the difference between our proposed method and existing work in Sec. 2, as suggested by **Reviewer Lood**.

* We have revised the figures; as suggested by **Reviewer UYRH**, we have enlarged the original figures and highlighted important regions for better visualization.

* We have included experiments compared with the single-person prediction baseline into Sec. G of the Appendix, as suggested by **Reviewer r2ic**.

Our revised text is marked in **teal**.

We again thank you for your valuable feedback and comments. We kindly remind you that we have provided additional results and clarifications to address your concerns based on your constructive comments. We would appreciate you being able to review our responses and let us know if you have any additional concerns or questions. And we are happy to clarify further.

Thanks!

---

### Author Response · Authors · 2022-11-28
**Please let us know if you have further questions**

Dear reviewers,

In case you didn’t notice, we have provided additional clarifications and paper revisions. We would like to know if our response resolves your questions. We are more than happy to discuss them if you have any further questions.

Thanks again for your constructive questions and feedback!

---

### Decision · Program_Chairs · 2023-01-20

**Decision:**

Accept: notable-top-25%

**Justification For Why Not Higher Score:**

For a paper like this, the quality of the predicted motion is important. Based on the supp materials, the results are good, but the detailed motion of humans is not captured.

**Justification For Why Not Lower Score:**

This is a solid submission and all four reviewers support acceptance (score: 8).

**Metareview: Summary, Strengths And Weaknesses:**

The submission studies the problem of stochastic human motion forecasting in a multi-human setting, which is a more challenging setup compared with most existing works. Overall, reviewers appreciated the clear motivation, the simple model, and the good results. There were concerns about the presentation, missing related work, and implementation details. The authors addressed the concerns nicely in the rebuttal. In the end, all four reviewers support acceptance. The AC agrees.

**Note From Pc:**

if the above contains the word "oral" or "spotlight" please see: "oral" presentation means -> notable-top-5% and "spotlight" means -> notable-top-25%. As stated in our emails, we are disassociating presentation type from AC recommendations